# SesaHand: Enhancing 3D Hand Reconstruction via Controllable Generation with Semantic and Structural Alignment

**Zhuoran Zhao**[1,2], **Xianghao Kong**[2], **Linlin Yang**[3], **Zheng Wei**[2], **Pan Hui**[1,2], **Anyi Rao**[2]
[1]The Hong Kong University of Science and Technology (Guangzhou)
[2]The Hong Kong University of Science and Technology
[3]Communication University of China
zzhao074@connect.hkust-gz.edu.cn anyirao@ust.hk

## Abstract

Recent studies on 3D hand reconstruction have demonstrated the effectiveness of synthetic training data to improve estimation performance. However, most methods rely on game engines to synthesize hand images, which often lack diversity in textures and environments, and fail to include crucial components like arms or interacting objects. Generative models are promising alternatives to generate diverse hand images, but still suffer from misalignment issues. In this paper, we present SesaHand, which enhances controllable hand image generation from both semantic and structural alignment perspectives for 3D hand reconstruction. Specifically, for semantic alignment, we propose a pipeline with Chain-of-Thought inference to extract human behavior semantics from image captions generated by the Vision-Language Model. This semantics suppresses human-irrelevant environmental details and ensures sufficient human-centric contexts for hand image generation. For structural alignment, we introduce hierarchical structural fusion to integrate structural information with different granularity for feature refinement to better align the hand and the overall human body in generated images. We further propose a hand structure attention enhancement method to efficiently enhance the model's attention on hand regions. Experiments demonstrate that our method not only outperforms prior work in generation performance but also improves 3D hand reconstruction with the generated hand images.

## 1 Introduction

3D hand reconstruction from a single image plays a crucial role in computer vision, human-computer interaction, and AR/VR applications (Han et al., 2022; Hosain et al., 2021; Arimatsu & Mori, 2020). It also has a great potential in dexterous robotic manipulation for embodied intelligence (Gavryushin et al., 2025; Luo et al., 2025). Current state-of-the-art methods (Xu et al., 2023; Moon & Lee, 2020; Huang et al., 2023) rely on large amounts of accurate ground-truth labels for training, but obtaining these labels is expensive and time-consuming. Therefore, synthetic data has gained growing interest in the computer vision community (Tesch et al., 2025; Black et al., 2023; Wood et al., 2021; Hasson et al., 2019). Synthetic hand datasets typically feature various hand poses and are rendered by game engines to enhance realism (Gao et al., 2022; Li et al., 2023). Several recent studies have explored training 3D hand reconstruction models with real and synthetic data, achieving better results than only training with real data (Li et al., 2023; Zhao et al., 2025). However, hand texture maps and backgrounds used to create synthetic hand datasets are limited, which affects the diversity of synthetic data. Hands are often placed in environments incompatible with the contextual **semantics**. Moreover, most synthetic hand datasets only show floating hands without arms and hardly feature hand-object interaction (see Fig. 1(a)), conflicting with the human body **structure** and failing to reflect real human behavior.

Diffusion models have shown impressive capability in generating realistic images from textual descriptions (Rombach et al., 2022; Gu et al., 2022). Texts are used as semantic controls to provide images with diverse settings. Several studies further explore controllable image generation with

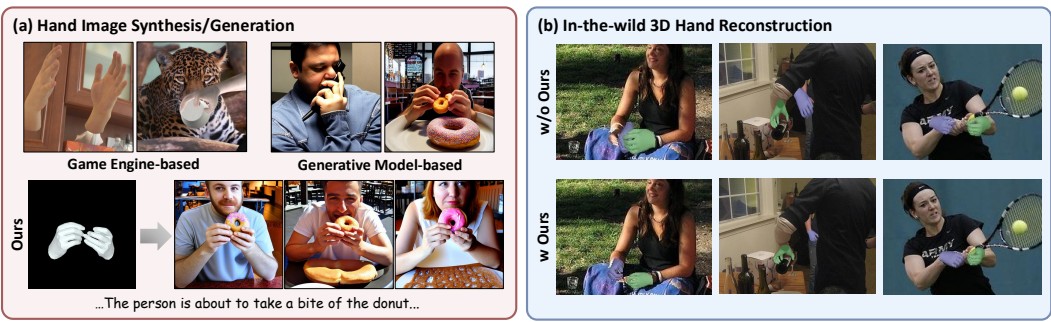

Figure 1: (a) We present a controllable hand image generation method that generates diverse hand images with semantic and structural alignment. (b) 3D hand reconstruction performance in the wild can be improved with better semantic- and structural- aligned generated images.

additional control modalities (Zhang et al., 2023a; Mou et al., 2024), which unlocks the potential for paired data generation. Pre-trained diffusion models gain rich human prior information (Chang et al., 2025; Kim et al., 2025) from massive training data, and controllable generative models ensure image-label alignment. These make diffusion models a good alternative to mitigate the limitations of existing hand image synthesis approaches.

Several works employ controllable diffusion models to generate hand data. However, they are restricted to lab environments (Xu et al., 2024), suffer from inefficient training for hand-related feature refinement (Park et al., 2024), or are skeleton-based, ignoring the influence of hand shape (Chen et al., 2025a). Moreover, prior work (Park et al., 2024) directly uses the large vision-language model (VLM) for image description generation, which poses challenges for plausible hand image generation due to the *overthinking* issues (Xiao & Gan, 2025) of VLMs. The overthinking issue can introduce irrelevant information into image descriptions, creating a **semantic** gap from insufficient human behavior context and leading to implausible hand image generation. Moreover, since the hand is a critical part of the human body, neglecting the **structural** information of the human body associated with the hand can cause misalignment issues in generated images, such as floating hands and implausible human poses (see Fig. 1(a)).

With these motivations in mind, we propose SesaHand, enhancing text-conditioned controllable hand image generation from both semantic and structural alignment perspectives. For **semantic** alignment, we find that the human-centric context within the image caption is crucial for generating plausible hand images and can be decomposed into four components: human pose, action, hand action, and the environment through prompt influence and attention analyses. We define this human-centric context as human behavior semantics. To extract this semantics effectively, we propose a pipeline with Chain-of-Thought (CoT) inference (Wei et al., 2022; Zhang et al., 2023b), which employs a step-by-step thinking approach to identify and extract human behavior semantics while eliminating irrelevant details. Specifically, the CoT inference consists of three stages. First, a Captioner generates an initial image caption for an input image. Second, an Extractor identifies and decomposes essential components of the image caption with few-shot learning. Third, a Composer composes the extracted components together into the final text prompt. The proposed CoT inference can generate image descriptions with sufficient semantic information about human behavior, facilitating the construction of improved text-image pairs to train the text-to-image model.

In diffusion-based models, self-attention maps preserve geometric and structural information of images, which has been proven useful for image editing (Liu et al., 2024a) and human image animation (Chang et al., 2025). This motivates us to strengthen the alignment of the hand and human body in controllable image generation by leveraging the **structural** information within self-attention maps. Specifically, we integrate structural information with different granularity with both global and local representations by extracting multi-resolution self-attention maps in the encoding and middle blocks of the control module (Zhang et al., 2023a) and hierarchically fuse them to refine the features fed into the Stable Diffusion generative backbone (Rombach et al., 2022). This helps provide enhanced structural information useful for hand image generation. Moreover, cross-attention maps reveal the semantic correspondence between image features and text embeddings. To high-

light the local hand structure in image features, we propose an efficient approach of adding bias terms to the hand-related cross-attention maps, rather than relying on a slow embedding refinement process (Park et al., 2024).

Extensive experiments demonstrate that our method outperforms existing approaches in hand image generation. Notably, we further show that the generated hand images improve the 3D hand reconstruction performance on in-the-wild datasets (see Fig. 1(b)), paving the path for exploring the use of synthetic data from generative models in 3D hand reconstruction task. We summarize our contributions as follows:

1. We propose a CoT inference-based pipeline that extracts human behavior semantics to construct text-image pairs for training, which mitigates overthinking issues in VLM-generated captions and improves semantic alignment in hand image generation.

2. We improve the structural alignment in hand image generation through a hierarchical structural fusion method that integrates multi-level features for hand-body alignment, along with a hand structure attention mechanism that efficiently highlights hand-related regions.

3. We demonstrate through extensive experiments that our method not only outperforms existing hand image generation methods, but also enhances in-the-wild 3D hand reconstruction performance using our generated images.

## 2 RELATED WORKS

**Hand Image Synthesis.** Creating real-world 3D hand datasets is both time-consuming and labor-intensive. Data collection typically involves using multi-camera or depth sensor setups (Chao et al., 2021; Zimmermann et al., 2019; Hampali et al., 2020; Zhan et al., 2024; Chen et al., 2025b). Manual labeling is required to obtain ground-truth annotations. Synthetic hand data becomes an effective alternative. Early methods synthesize realistic hand images with game engines, which can render large-scale synthetic data efficiently (Gao et al., 2022; Moon et al., 2023; Li et al., 2023; Hasson et al., 2019). However, these images often show floating hands without arms, which is unnatural from the human behavior perspective, and incorporating human bodies requires an extra fitting process. Also, synthesizing realistic images requires additional efforts to collect hand texture maps and environment backgrounds, which are often limited in quantity and are not customizable (Li et al., 2023; Moon et al., 2023). Our method generates hand images with natural human bodies and uses text prompts to provide diverse textures and environments. Recent works explore hand image generation with diffusion models. HanDiffuser (Narasimhaswamy et al., 2024) and Hand1000 (Zhang et al., 2025) generate hand images from text prompts directly, but the generated images cannot be used to train 3D hand reconstruction models since they have no reliable corresponding labels. FoundHand (Chen et al., 2025a) achieves controllable hand image generation from 2D keypoints, which ignores the hand shape factor. HandBooster (Xu et al., 2024) supports hand image generation with various conditions, but it is limited to lab environments. Similar to our method, Attention-Hand (Park et al., 2024) generates in-the-wild hand images with hand mesh conditions and text prompts. However, it suffers from a huge training time cost in optimizing hand-related features and two-stage learning for global and local features. Our work achieves better generation performance without sacrificing time cost, and the generated hand images are useful for improving 3D hand reconstruction performance in in-the-wild scenarios.

**Text-to-Image Generation.** Text-to-Image (T2I) generation has seen significant progress in recent years, enabling diverse and realistic image generation based on text descriptions. Early approaches utilize the generative adversarial network (GAN) (Reed et al., 2016; Xu et al., 2018), and transformers also show potential for conditional generation (Ding et al., 2021; Yu et al., 2022). The emergence of diffusion models has marked a new era for T2I generation (Rombach et al., 2022; Saharia et al., 2022; Chen et al., 2024). Since text prompts cannot provide enough structural guidance, recent research introduces additional control modalities to enable customized and controllable T2I generation. ControlNet (Zhang et al., 2023a) integrates user-specific conditional information into the image generation process by creating a trainable copy of the Stable Diffusion model. T2I-Adapter (Mou et al., 2024) also adapts Stable Diffusion to external control modalities with trainable adapters. Despite these advances, challenges still remain in hand image generation due to the small size of the hand and ambiguous occlusion problems. In this work, we enhance hand region learning

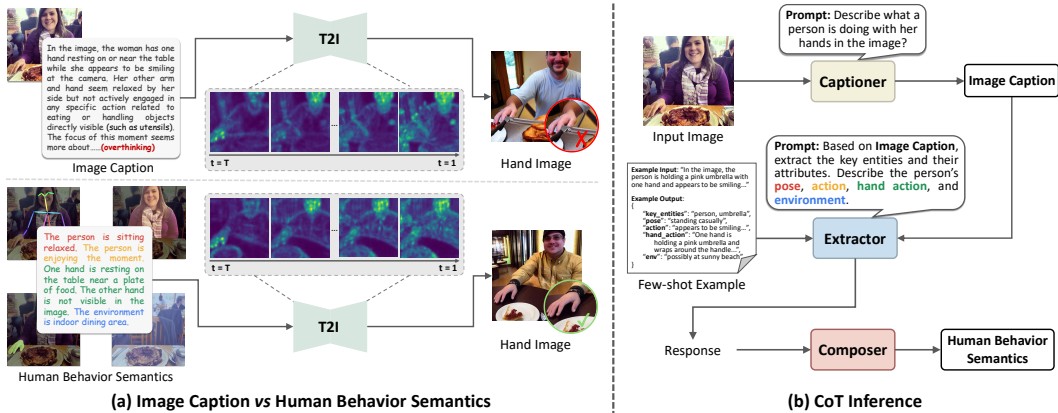

Figure 2: (a) Comparison of hand image generation with VLM-generated caption (top) and human behavior semantics (bottom). Overthinking in VLM captions leads to attention shifts toward irrelevant objects in later denoising steps, while human behavior semantics guide the model to focus on human-related regions, generating more plausible hand images. (b) CoT inference in human behavior semantics extraction pipeline.

through hand structure attention enhancement and reduce ambiguity by extracting human behavior semantics from image descriptions for generation.

## 3 METHOD

Our method aims to enhance text-conditioned hand mesh image-controlled image generation with both semantic and structural alignment. In the preliminary, we provide an overview of ControlNet (Sec. 3.1), a control module for Stable Diffusion, which facilitates controllable hand image generation based on the hand mesh image. Firstly, we introduce our human behavior semantics extraction pipeline with Chain-of-Thought (CoT) inference that incorporates crucial components to achieve semantic alignment for hand image generation (Sec. 3.2). Secondly, we demonstrate our hierarchical structural fusion method to improve the hand-body alignment (Sec. 3.3.1) and our hand structure attention enhancement method to emphasize local hand regions efficiently (Sec. 3.3.2).

### 3.1 PRELIMINARY

Latent diffusion model (Rombach et al., 2022) is an image generation method that performs the denoising process in the image latent space. It utilizes a pretrained image encoder to encode the input image into its latent embedding: $z = \mathcal{E}(x)$. During training, the model optimizes a UNet (Ronneberger et al., 2015) $\epsilon_\theta$ to predict the noise added to a noisy latent sample $z_t$ conditioned on the text prompt embedding $c_t$. Here, $z_t$ is a noisy sample of $z_0$ at a random step $t \in [0, T]$. At inference time, the model generates desired samples from the image latent space, starting from Gaussian noise $z_T \sim \mathcal{N}(0, 1)$ and denoising through $T$ timesteps.

ControlNet (Zhang et al., 2023a) adds conditional control to a pretrained diffusion model, such as Stable Diffusion. It preserves the original model's capabilities by locking its parameters and creating a trainable copy of its encoding blocks, which is connected to the locked model via zero convolution layers $\mathcal{Z}(.;.)$ to stabilize training in initial steps. The training objective of ControlNet is given by:

$$L(\theta) := \mathbb{E}_{z_0, \epsilon \in \mathcal{N}(0,1), t, c_t, c_f} \left[ \|(\epsilon - \epsilon_\theta(z_t, t, c_t, c_f))\|_2^2 \right], \tag{1}$$

where $c_f$ is a task-specific latent condition encoded by a small neural network from input condition $c_i$ in image space: $c_f = \mathcal{E}_n(c_i)$. In our case, $c_i$ denotes the hand mesh image and $c_f$ denotes its latent representation.

## 3.2 Semantic Alignment: Human Behavior Semantics Extraction

To achieve plausible hand image generation, it is essential to include sufficient semantic information about human behavior in text prompts while eliminating irrelevant details of other elements. VLMs are effective for image captioning (Lin et al., 2025; Ge et al., 2024) and can be used to efficiently collect paired text-image data for training T2I models. However, VLMs exhibit *overthinking* issues (Xiao & Gan, 2025), which tend to describe every element in images in detail regardless of the specific scenario. Such irrelevant information about non-human elements can introduce bias to the generative model and degrade the generated hand image quality (Park et al., 2024). As shown in Fig. 2(a), the image description generated by VLM (Bai et al., 2025) contains redundant descriptions such as utensils, which causes unnecessary occlusions of the hand in the generated image.

Although overthinking issues exist, we find that the human-centric context in image caption is crucial for generating plausible hand images. A hand image can be semantically decomposed into human and environment components. To better capture human-centric information, we further decompose the human component into three finer-grained parts: human pose, action, and hand action (see Fig. 2(a)). We define this human-centric context as human behavior semantics, which represent essential elements within a hand image while eliminating irrelevant details that bias the T2I generation process. To verify the superiority of the proposed human behavior semantics, we perform a prompt influence analysis by first generating captions and extracting human behavior semantics on 100 image samples. We then use either captions or human behavior semantics as text prompts along with hand mesh images to generate hand images and compute average hand confidence scores with a hand detector (Zhang et al., 2020). Hand images generated with human behavior semantics achieve a higher confidence score (97%) compared to VLM captions (86%). Moreover, we analyze attention maps in the T2I model to uncover the generation process (see Appendix A). We observe that overthinking in VLM caption causes model's attention to deviate towards irrelevant object generation, especially in later denoising timesteps. In contrast, the T2I model with human behavior semantics generates a more plausible hand image by preserving focused attention on human-related regions.

Based on this observation, we propose a pipeline with CoT inference to extract human behavior semantics effectively (see Fig. 2(b)). CoT has been validated as an effective means of guiding the reasoning process of LLMs (Wei et al., 2022). Therefore, we introduce CoT to our pipeline to generate human behavior semantics utilizing image caption and critical decomposed components with few-shot learning. Specifically, for an image X, its associated image description $X_t$ is first generated by Captioner: $X_t = \text{Captioner}(X)$. Then we use Extractor to extract the human behavior semantics within the image given $X_t$: $P = \text{Extractor}(X_t, P_e)$, where $P_e$ is a few-shot example of input and output that guides Extractor to extract key entities and attributes more accurately. We restrict the ouput to the JSON format since it has better parsability, enabling easier interpretations (Mitra et al., 2024). Then we employ a Composer to extract the components from the output JSON and compose them together to obtain the final text $P_f$:

$$P_f = \text{Composer}\left(P_{\text{pose}}, P_{\text{action}}, P_{\text{hand\_action}}, P_{\text{env}}\right). \qquad (2)$$

The final composed texts $P_f$ are subsequently used to construct text-image pairs for training our controllable hand image generation model.

## 3.3 Structural Alignment

### 3.3.1 Hierarchical Structural Fusion

To strengthen the hand-body alignment in hand image generation, we propose a multi-level structural information fusion approach. We adopt Stable Diffusion as the generative backbone and utilize ControlNet to incorporate 2D hand mesh control. ControlNet preserves the capability of the pre-trained Stable Diffusion model by locking its parameters while creating a trainable copy of its encoding and middle blocks. The encoding and middle blocks consist of ResNet layers and transformer blocks, which generate features of different resolutions (64, 32, 16, 8). Each transformer block consists of self- and cross-attention layers. We extract the self-attention maps generated from $z_t$ and $c_f$, denoted $\psi_r$, where $r$ represents the resolution. Self-attention maps with higher resolution capture fine-grained local human structural information, while lower resolution maps focus on the global human structural information (Chu et al., 2017).

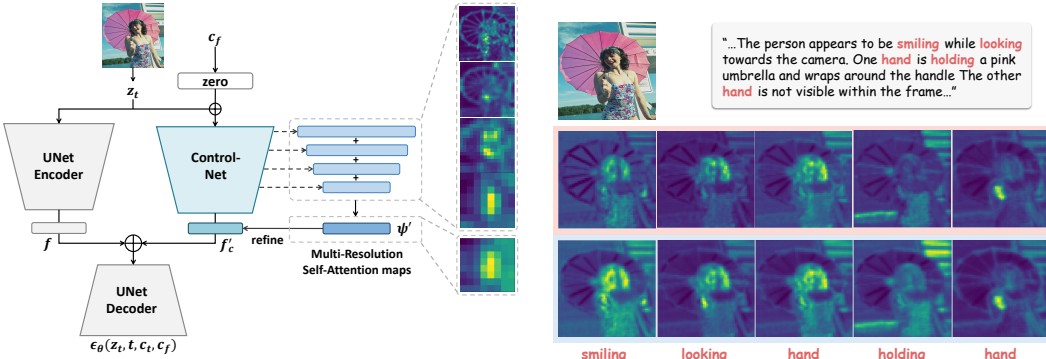

Figure 3: Hierarchical Structural Fusion. Multi-level self-attention maps are extracted from the ControlNet encoder and middle blocks, which capture the structural information of the input image. These maps are aggregated and applied to obtain the refined feature fed to the Decoder.

Figure 4: Hand Structure Attention Enhancement. Applying the enhancement (bottom) effectively highlights the local structural human- and hand-related features compared to the original cross-attention maps (top).

As shown in Fig. 3, the noisy image latent $z_t$ is first combined with the conditioning hand mesh feature $c_f$ and then input to the control module. Self-attention maps in encoding and middle blocks are extracted to refine the original feature $f_c$ output by the control module. Specifically, self-attention maps in different resolutions are first resized to the same resolution by applying the max pooling operation $\mathcal{M}$ and then aggregated by summation:

$$\psi' = \sum_{r=8,16,32,64} \mathcal{M}(\psi_r). \tag{3}$$

Then the aggregated self-attention map $\psi'$ is applied to the $f_c$ to produce the refined feature $f_c'$:

$$f_c' = f_c \otimes \psi', \tag{4}$$

where $\otimes$ denotes matrix multiplication.

The refined feature is integrated into the original output $f$ of the Stable Diffusion model through zero convolution layers $\mathcal{Z}$:

$$f' = \mathcal{Z}(f_c') + f. \tag{5}$$

This refined feature preserves more human structural information, which helps the UNet Decoder to generate hand images with improved structural alignment.

### 3.3.2 HAND STRUCTURE ATTENTION ENHANCEMENT

To emphasize the local structural features of the hand region in images efficiently, we introduce our hand structure attention enhancement approach. The cross-attention layers in the Stable Diffusion model inject textual information into the image generation process, enabling T2I models to generate images consistent with the text prompts. To enhance the generation of hand-related regions, we first perform hand-related tagging on text prompts using part-of-speech tagging (Chiche & Yitagesu, 2022) from NLTK library (Loper & Bird, 2002) following (Park et al., 2024). We extract the index list $I$ of input tokens belonging to the "Verb" category and containing the word "hand". We then derive the hand-related features through the cross-attention layers in the control module with the extracted index list $I$. Instead of refining noisy embedding $z_t$ through a time-consuming optimization process (Park et al., 2024), we enhance hand-related features by applying a bias term to specific cross-attention maps, which enables hand-related attention maps to contribute more to the final feature in attention-weighted combination, improving hand image generation. In the cross-attention layer of the control module, the query $Q_i$ is derived from $\phi_i(z_t, c_f)$, where $z_t$ is the noisy image embedding, $c_f$ is the latent hand mesh conditioning image, and $\phi_i(z_t, c_f)$ is the input of the $i$-th cross-attention layer. The keys $K_i$ and values $V_i$ are derived from the text embeddings $c_t$ produced by the CLIP text encoder (Radford et al., 2021). The cross-attention maps $M_{cross}$ are computed as:

$$M_{cross} = \text{softmax}(\frac{Q_i K_i^T}{\sqrt{d_i}} + B), \tag{6}$$

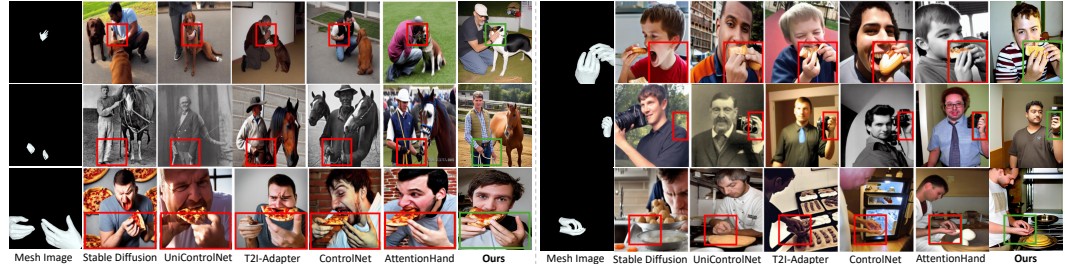

Figure 5: Qualitative comparisons with state-of-the-art text-to-image generation models.

Table 1: Quantitative comparisons on MSCOCO.

| Method | FID ↓ | KID ↓ | FID-H ↓ | KID-H ↓ | Hand Conf. ↑ | MSE-2D ↓ | MSE-3D ↓ | User Pref. ↑ |
|---|---|---|---|---|---|---|---|---|
| Stable Diffusion (Rombach et al., 2022) | 40.52 | 0.00684 | 50.78 | 0.02554 | 0.651 | 2.932 | 4.591 | 1.50 |
| Uni-ControlNet (Zhao et al., 2023) | 30.34 | 0.00744 | 37.77 | 0.02004 | 0.855 | 2.105 | 3.039 | 2.56 |
| T2I-Adapter (Mou et al., 2024) | 22.00 | 0.00761 | 32.08 | 0.01568 | 0.914 | 1.546 | 2.451 | 2.91 |
| ControlNet (Zhang et al., 2023a) | 21.67 | 0.00658 | 40.32 | 0.02098 | 0.810 | 1.252 | 2.182 | 3.21 |
| AttentionHand (Park et al., 2024) | 20.71 | **0.00301** | 27.09 | 0.01287 | 0.965 | 1.026 | 1.986 | 4.05 |
| **Ours** | **18.63** | 0.00553 | **17.77** | **0.00718** | **0.966** | **1.015** | **1.537** | **4.55** |

where $d_i$ is the dimension of the key, $B$ is a bias matrix introduced to enhance the hand-related feature to improve the generation of hand regions. The bias matrix $B$ is defined as:

$$B_{q,k} = \begin{cases} \alpha, & k \in I \\ 0, & \text{otherwise} \end{cases}, \qquad (7)$$

where $\alpha$ is a positive hyperparameter, $q$ is the query index and $k$ is the key index.

The output of the cross-attention layer is computed by applying the enhanced cross-attention maps $M_{cross}$ to the value $V_i$:

$$\phi_i'(z_t, c_f) = M_{cross} V_i. \qquad (8)$$

As shown in Fig. 4, this approach enhances the model's ability to capture and emphasize hand-related features, thus improving the generation of hand regions.

## 4 EXPERIMENTS

### 4.1 EXPERIMENTAL SETTING

**Datasets.** For text-to-image generation, we follow prior work (Park et al., 2024) to use MSCOCO (Lin et al., 2014; Jin et al., 2020), which is preprocessed with RGB hand and hand mesh images for training and evaluation. We generate new image descriptions with our human behavior semantics extraction pipeline. For 3D hand mesh reconstruction, we evaluate on two in-the-wild test datasets Hands-In-Action (HIC) (Tzionas et al., 2016) and Re:InterHand (ReIH) (Moon et al., 2023). HIC consists of single and interacting hand sequences recorded with an RGBD camera with 732 samples in the test set. ReIH is a synthetic interacting hand dataset, which is relighted with HDR backgrounds. ReIH test set has 126,640 samples. We show our qualitative results on MSCOCO.

**Evaluation Metrics.** To evaluate image generation performance, we adopt Frechet Inception Distance (FID) (Heusel et al., 2017) and Kernel Inception Distance (KID) (Bińkowski et al., 2018). We evaluate the quality of generated images in the hand regions with FID-H and KID-H by computing FID and KID on hand crops. We also evaluate the hand confidence score (Hand Conf.) to measure the hand quality in generated images with an off-the-shelf hand detector Mediapipe (Zhang et al., 2020) to compute the average confidence scores for detection following (Park et al., 2024; Narasimhaswamy et al., 2024). We follow prior work (Park et al., 2024) to report mean squared error of 2D keypoints (MSE-2D) with Mediapipe and mean squared error of 3D keypoints (MSE-3D) using InterWild (Moon, 2023). For user preference (User Pref.), we asked 12 volunteers to rate 10 groups of generate hand images in the following aspects with a 5-point Likert scale: visual quality, hand mesh alignment, and text alignment. To evaluate 3D hand reconstruction performance,

Table 2: Quantitative comparisons of 3D hand reconstruction methods with and without our generated images. * indicates our re-implemented results.

| Datasets Method | HIC (Tzionas et al., 2016) | | | ReIH (Moon et al., 2023) | | |
|---|---|---|---|---|---|---|
| | MPVPE↓ | RRVE↓ | MRRPE↓ | MPVPE↓ | RRVE↓ | MRRPE↓ |
| DIR (Ren et al., 2023) | 21.89 | 26.11 | 43.11 | 21.82 | 29.66 | 37.01 |
| + AttentionHand | 20.66 $_{(-5.6\%)}$ | 25.87 $_{(-0.9\%)}$ | 40.54 $_{(-6.0\%)}$ | 19.91 $_{(-8.8\%)}$ | **26.67** $_{(-\mathbf{10.1\%})}$ | 35.05 $_{(-5.3\%)}$ |
| DIR* (Ren et al., 2023) | 21.94 | 29.09 | 42.60 | 22.14 | 29.41 | 40.48 |
| + Ours | **20.48** $_{(-\mathbf{6.7\%})}$ | **27.78** $_{(-\mathbf{4.5\%})}$ | **40.01** $_{(-\mathbf{6.1\%})}$ | **19.21** $_{(-\mathbf{13.2\%})}$ | 27.42 $_{(-6.8\%)}$ | **38.10** $_{(-\mathbf{5.9\%})}$ |
| InterWild (Moon, 2023) | 15.30 | 21.35 | 31.26 | 13.99 | 20.07 | 22.38 |
| + AttentionHand | 14.74 $_{(-3.7\%)}$ | 21.10 $_{(-1.2\%)}$ | 29.26 $_{(-6.4\%)}$ | 13.95 $_{(-0.3\%)}$ | 19.94 $_{(-0.6\%)}$ | **22.05** $_{(-\mathbf{1.5\%})}$ |
| InterWild* (Moon, 2023) | 15.30 | 24.20 | 31.26 | 13.99 | 20.07 | 22.38 |
| + Ours | **14.70** $_{(-\mathbf{3.9\%})}$ | **23.36** $_{(-\mathbf{3.5\%})}$ | **28.57** $_{(-\mathbf{8.6\%})}$ | **13.01** $_{(-\mathbf{7.0\%})}$ | **19.32** $_{(-\mathbf{3.7\%})}$ | 22.17 $_{(-0.9\%)}$ |

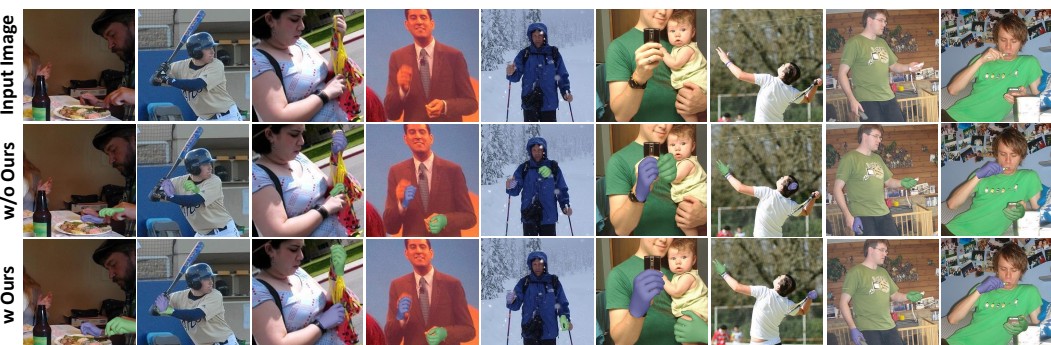

Figure 6: Qualitative comparison of InterWild with and without our generated images on MSCOCO validation set.

we report three metrics: mean per-vertex position error (MPVPE), right hand-relative vertex error (RRVE), and mean relative-root position error (MRRPE) in mm. See Appendix C for more details.

## 4.2 MAIN RESULTS

**Comparison on Image Generation.** We compare our method with state-of-the-art controllable image generation methods. As shown in Tab. 1, our method outperforms other methods on FID, FID-H, KID-H, and hand confidence score. Specifically, our method achieves improvements of 34% of FID-H and 44% of KID-H over AttentionHand, demonstrating the effectiveness of introducing semantic and structural information to improve hand image generation. Also, AttentionHand directly resizes images for training, which can result in a hand distortion problem that affects image generation performance. We further perform a qualitative comparison to visually evaluate the effectiveness of our method (see Fig. 5). Our method achieves high-fidelity generation results compared with other methods. The hands in the generated images are aligned with the hand mesh image conditions, and the associated human bodies are in plausible poses. The results also show that the model can generate human bodies compatible with varying hand poses and shapes.

**Comparison on 3D Hand Reconstruction.** We evaluate the effectiveness of our method by fine-tuning two 3D hand reconstruction methods InterWild (Moon, 2023) and DIR (Ren et al., 2023) with our generated images on HIC and ReIH. As shown in Tab. 2, our method outperforms AttentionHand, especially in MPVPE, achieving an improvement of 3.9% v.s 3.7% on HIC and 7.0% v.s 0.3% on ReIH with InterWild, and 6.7% v.s 5.6% on HIC and 13.2% v.s 8.8% on ReIH with DIR. We also show qualitative results in Fig. 6. Although InterWild is trained with large-scale real and synthetic datasets, it still underperforms on challenging in-the-wild samples, such as those involving occlusions and truncations. Our generated images can enhance the model's robustness in hand detection as well as hand mesh predictions.

Table 3: Ablation study of different components. SE, SF, and AE denote semantics extraction, structural fusion, and attention enhancement. HC denotes the hand confidence score.

| SE | SF | AE | FID ↓ | KID ↓ | FID-H ↓ | KID-H ↓ | HC ↑ |
|---|---|---|---|---|---|---|---|
| ✗ | ✗ | ✗ | 21.04 | 0.00744 | 21.90 | 0.00982 | 0.942 |
| ✓ | ✗ | ✗ | 19.83 | 0.00664 | 18.90 | 0.00807 | 0.944 |
| ✓ | ✓ | ✗ | 19.05 | **0.00542** | 18.17 | 0.00722 | 0.960 |
| ✓ | ✓ | ✓ | **18.63** | 0.00553 | **17.77** | **0.00718** | **0.966** |

Table 4: Speed comparison.

| Method | ControlNet | AttentionHand | Ours |
|---|---|---|---|
| Duration (s/iter) | **0.41** | 27.25 | 0.44 |

Table 5: Comparison on different bias $\alpha$.

| $\alpha$ | 1.0 | 1.5 | 2.0 | 2.5 |
|---|---|---|---|---|
| FID ↓ | 20.41 | 19.69 | **18.63** | 19.97 |
| FID-H ↓ | 19.75 | 18.98 | **17.77** | 19.01 |

## 4.3 ABLATION STUDIES

**Component Ablations.** As shown in Tab. 3, we conduct an ablation study of our method components on generation performance. By incorporating human behavior semantics extraction pipeline, the performance can be improved from FID 21.04 to 19.83, demonstrating the effectiveness of human behavior semantics integration for improving image generation performance. We then introduce the hierarchical structural fusion and hand-related attention enhancement incrementally to improve structural alignment and highlight hand-related local features. Each addition further improves model performance. These experiments validate the contribution of each component to the overall performance. We further show visualizations of ablation studies (see Fig. 7).

We show two challenging samples: one real image with motion blur and occlusion issues (both self- and object-occlusion), and another with varying hand shape scales. The results show that adding human behavior semantics extraction improves the hand image quality since it highlights human- and hand-related context while eliminating irrelevant information. Hierarchical structural fusion injects the structural information of the human body into the generation process, facilitating the hand-body alignment. Furthermore, hand-related attention enhancement enables better generation of hand regions.

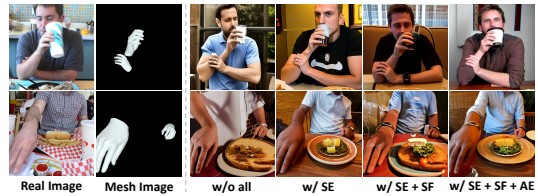

Real Image   Mesh Image   w/o all   w/ SE   w/ SE + SF   w/ SE + SF + AE

Figure 7: Ablation examples. SE, SF, and AE denote semantics extraction, structural fusion, and attention enhancement.

**Training Efficiency Analysis.** We perform a training efficiency analysis compared with ControlNet and AttentionHand. As shown in Tab. 4, although our training speed is slightly slower than ControlNet, we are much faster than AttentionHand, which requires a complex optimization process during training for hand-related feature refinements.

**Ablations on the Bias Term.** We conduct a hyperparameter search on the $\alpha$ of the bias matrix to determine its value. As shown in Tab. 5, a bias value of 2.0 yields the best performance while other values have decreased results. This indicates that small bias enhancement may fail to highlight the useful hand-related regions while high bias introduces noise.

## 4.4 HUMAN PERCEPTUAL STUDY

We further evaluate SesaHand with two human perceptual studies. The first is to evaluate our human behavior semantics extraction pipeline compared to different VLMs, including Qwen-VL (Bai et al., 2023), Qwen2.5-VL-7B (Bai et al., 2025), and LLaVA-v1.6-7B (Liu et al., 2024b). We randomly select 12 images and their descriptions generated by the VLMs and our pipeline, and conduct our study with 20 users. Users are asked to select the best description of human behavior in images among four descriptions. We collect 240 answers and report the corresponding preference rate. Fig. 8(a) shows that our pipeline outperforms other VLMs, achieving a 41% preference rate. The second

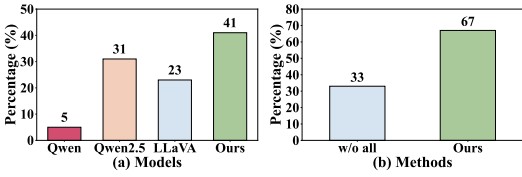

Figure 8: Human perceptual studies.

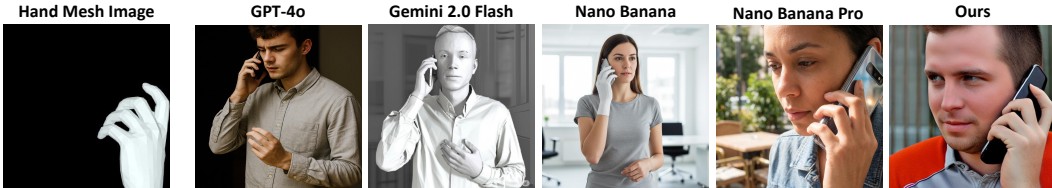

Figure 9: Comparison with commercial models. Given a conditioning hand mesh image and a text prompt, GPT-4o, Gemini 2.0 Flash, Nano Banana, and Nano Banana Pro fail to generate well-aligned and realistic hand images, despite their capability for multiple image generation tasks.

is to compare images generated by the model without our improvements to those generated with all improvements applied. We present 12 groups of generated images with text prompts and hand mesh conditions. Users are asked to select the best image regarding text alignment, structural control, and image quality. Our preference rate is 67% (see Fig. 8(b)), indicating the effectiveness of our approach. See Appendix B for more details.

### 4.5 COMPARISON WITH COMMERCIAL MODELS.

We further explore the ability of existing commercial models to generate well-aligned hand images. For qualitative comparison, we choose four widely-used commercial image generation models: GPT-4o (Hurst et al., 2024), Gemini 2.0 Flash (Comanici et al., 2025), Nano Banana (Gemini 2.5 Flash), and Nano Banana Pro (Gemini 3 Pro Image). As shown in Fig. 9, despite providing these four models with additional explanations about the task, these models fail to follow the instructions to generate aligned hand images. GPT-4o generates a realistic image, but it does not align with the conditioning hand mesh image. Gemini 2.0 Flash and Nano Banana are affected by the conditioning image and generate an undesired 3D rendering without texture. Nano Banana Pro performs better in generating realistic images, but still suffers from a misalignment problem with the conditioning mesh image. In contrast, our method can generate a realistic and well-aligned hand image.

## 5 CONCLUSION

We propose SesaHand, a controllable hand image generation method that achieves semantic and structural alignment to enhance 3D hand reconstruction. Previous hand image synthesis methods lack diversity and struggle to integrate critical components, while diffusion-based hand image generation methods suffer from misalignment and training inefficiency issues. For semantic alignment, we propose a human behavior semantics extraction pipeline with CoT inference to identify and generate crucial semantics, which helps extract essential human-centric context and reduces irrelevant information for better hand image generation. For structural alignment, we perform hierarchical structural fusion to refine the image features to enhance the hand-body alignment in the generated images. We also propose a hand structure attention enhancement method that emphasizes the local hand features to improve the generation of hand regions. With these improvements, our method outperforms state-of-the-art methods in both quantitative and qualitative evaluations.

## 6 LIMITATION AND FUTURE WORK

Our method may generate blurry hand images when the conditioning hand is small. When hand-object interaction exists, the hand and object sometimes fuse with each other at contact points. In future work, we plan to enhance the learning of the hand-object interaction region to achieve more realistic and natural hand-object interactions in generated images. Also, adopting a prioritized training strategy that emphasizes higher noise levels training with a higher sampling probability could further improve hand structure generation quality, since higher noise levels focus on generating images' structural information while lower noise levels focus on refining details. Finally, incorporating additional paired training data from ego-centric datasets can enhance our method's ability to generate better ego-centric images, benefiting downstream tasks such as robotic manipulation.

## ACKNOWLEDGEMENT

This research was supported in part by the NSFC/RGC Collaborative Research Scheme Project No.CRS_HKUST605/25, the National Natural Science Foundation of China (No.62561160115), the National Key Research and Development Program of China under Grant 2024YFC3307602, and the Guangdong Provincial Talent Program, Grant No.2023JC10X009. Additionally, this research was supported by National Natural Science Foundation of China (No.62406298).

## ETHICS STATEMENT

This work adheres to the ICLR Code of Ethics. All user data involved in the human perceptual study was rigorously anonymized. The datasets used in our research were accessed from published literature and open-source platforms such as *GitHub* and *Hugging Face*. Our work aims to improve the hand image generation from a technical perspective and is not intended for malicious use.

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

APPENDIX OVERVIEW

The appendix includes human behavior semantics extraction pipeline details, human perceptual study details, implementation details, additional quantitative and qualitative comparisons. The appendix is organized as follows:

- Sec. A provides our attention analysis and the few-shot example used in our human behavior semantics extraction pipeline.

- Sec. B shows more details of our human perceptual study.

- Sec. C provides the implementation details.

- Sec. D presents the module-wise runtime analysis.

- Sec. E shows more qualitative results and comparisons with FoundHand (Chen et al., 2025a), HandBooster (Xu et al., 2024), ControlNet (Zhang et al., 2023a) and Attention-Hand (Park et al., 2024).

## A   HUMAN BEHAVIOR SEMANTICS EXTRACTION DETAILS

### A.1   ATTENTION ANALYSIS

In this section, we show our attention analysis of the Stable Diffusion model with VLM-generated image caption and human behavior semantics. As shown in Fig. 10, we visualize the attention maps in the UNet Decoder during different timesteps. We choose an early block and a later block of the UNet Decoder as examples. In initial timesteps, both attention maps from the VLM-generated image caption and human behavior semantics focus on the generation of human structure and the hand region. As the timestep increases, the model starts to add more irrelevant environmental details depicted in the prompt with VLM caption, which severely occlude the hand. In contrast, human behavior semantics enables the model to refine details while still preserving the hand region. This observation indicates that overthinking in image caption can cause attention deviation towards generating those irrelevant objects that degrade the hand image generation. Human behavior semantics with essential components mitigate this deviation and improve hand image generation performance.

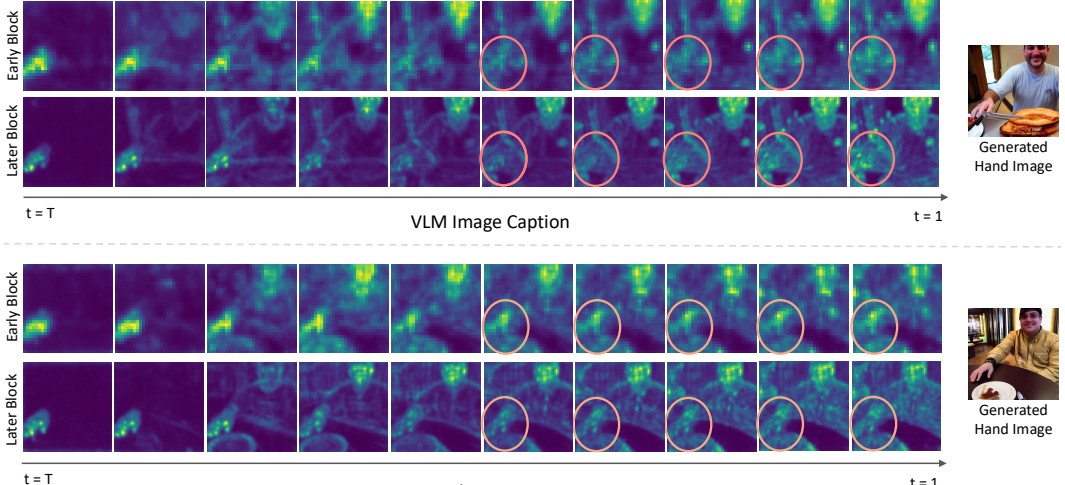

Figure 10: Visualization of attention maps in UNet Decoder early and later blocks with VLM-generated image caption (top) and human behavior semantics (bottom). VLM-generated caption causes attention deviation towards irrelevant objects in later denoising steps, while human behavior semantics enables more focused attention on human- and hand-related regions.

## A.2 FEW-SHOT EXAMPLE

In human behavior semantics extraction pipeline, we provide a few-shot example to guide Extractor to generate more accurate responses that conform to the JSON format for better composition. Here is the few-shot example:

**Example Input:** "In the image, the person is holding a pink umbrella with one hand and appears to be smiling while looking towards the camera..."

**Example Output:**
{
  **"key_entities":** "person, umbrella",
  **"pose":** "standing casually",
  **"action":** "appears to be smiling while looking towards the camera",
  **"hand_action":** "One hand is holding a pink umbrella and wraps around the handle...",
  **"env":** "possibly at sunny beach"
}

## B   HUMAN PERCEPTUAL STUDY

We recruited 20 users via social media, including gender (12 female and 8 male) and age ($M = 26.33$, $SD = 1.24$). 7 users are AI researchers, 3 are HCI researchers, 3 have technical backgrounds, and 7 are from other fields. We design a questionnaire and share it with each user. There are 12 groups of questions, and we collected 240 answers in total. The preference rate for each model is calculated as its number of selections divided by the total number of answers.

In the first human perceptual study, we ask each user to select the description that best reflects human behavior for each image, based on the following question and background information:

**Question:** "Which image description most accurately and clearly describes the human behavior without hallucination?"

**Human behavior** refers to the range of actions, reactions, and interactions exhibited by people in response to their environment.

**Hallucination** refers to descriptions that are irrelevant or redundant for understanding human behavior.

As shown in Fig. 8 of the main paper, our visual prompt generation pipeline outperforms state-of-the-art vision-language models in describing human behavior within the image. Our visual prompt generation pipeline achieves a 41% preference rate, surpassing the second-best model Qwen2.5-VL-7B (Bai et al., 2025) with a 31% preference rate. LLaVA-v1.6-7B and Qwen-VL have 23% and 5% preference rates, respectively. We observe that Qwen-VL (Bai et al., 2023) tends to generate brief image descriptions and cannot provide detailed information about human behavior, likely due to its limited visual reasoning ability. Qwen2.5-VL-7B is stronger than Qwen-VL, but exhibits overthinking issues and produces descriptions unrelated to human behavior. LLaVA-v1.6-7B (Liu et al., 2024b) performs well in some cases, but still suffers from hallucination and struggles to describe human behavior accurately.

The second human perceptual study is about image generation. We sample 12 images generated by the model without our improvements (baseline) and the model with our improvements applied. Users are asked to select the better image in each group based on three criteria: text alignment, hand alignment, and generation quality:

**Question:** "Which image better matches the following text prompt, aligns with the hand mesh image condition, and has better image quality?"

A total of 240 answers are collected and the preference rate is calculated using the same method as in the first human perceptual study. As shown in Fig. 8 of the main paper, our method achieves superior performance in human preferences with a 67% preference rate, further validating its effectiveness. During the interview, users reported that images generated by the baseline model exhibit mismatched hand joints, unnatural hand-object interactions (with the hand and object fusing at contact points), and incorrect hand shapes. Moreover, the baseline model fails to handle perspective well, which results in an improper size of the arm relative to the hand in the generated images.

## C    IMPLEMENTATION DETAILS

For CoT inference-based human behavior semantics extraction, we use Qwen2.5-VL (Bai et al., 2025) as Captioner and LLaMA 3 70B (Grattafiori et al., 2024) as Extractor for semantics extraction. For Composer, we integrate different output components together using a concatenation operation.

We use Stable Diffusion 1.5 (Rombach et al., 2022) as our generative backbone and freeze its weight during training. An AdamW optimizer is utilized with a learning rate of 1e-5 to train the control module for 50 epochs with a batch size of 2. The hyperparameter $\alpha$ is set to 2.0. Experiments are conducted on one NVIDIA RTX 5000 Ada. The training takes around 2 days (50 GPU hours). We crop the input RGB image and the hand mesh image in the training set centered on the two-hands region and resize it to $512 \times 512$ for training to prevent distortion of the hand and human body caused by direct resizing.

For the evaluation of 3D hand reconstruction, we reimplement InterWild (Moon, 2023) and DIR (Ren et al., 2023) using their official code. For InterWild, we select the checkpoint trained with InterHand2.6M (H+M) (Moon et al., 2020), MSCOCO (Lin et al., 2014), and Re:InterHand (Ego_cameras) (Moon et al., 2023), since it produces results most similar to those reported in the AttentionHand paper, which does not provide evaluation implementation details. For DIR, we use the checkpoint released in its *GitHub* repository. For DIR evaluation, we preprocess the Re:InterHand test set by first cropping the hand region in each image based on 2D joints, following the data preprocessing procedure used in the original implementation of DIR.

For evaluation metrics, MPVPE calculates the Euclidean distance between predicted and ground-truth 3D hand meshes after aligning the translation of the right and left hand separately. RRVE computes the relative position between two hands, aligning the translation of the right hand's root joint. MRRPE measures the 3D relative distance between two hands, which is the 3D distance between the predicted and ground-truth right hand root-relative left hand root position.

## D    MODULE-WISE RUNTIME ANALYSIS.

We report module-wise runtime analysis in Tab. 6. Both Captioner and Extractor are deployed locally. The GPU memory usage is 26 GB during training and 17 GB during inference. Training and inference are both conducted on a single NVIDIA RTX 5000 Ada.

Table 6: Module-wise Runtime Analysis.

| Module | Captioner | Extractor | Attention Fusion | Diffusion Sampling | Inference |
|---|---|---|---|---|---|
| Runtime (s) | 1.7 | 0.8 | $2.9\times10^{-4}$ | 0.44 | 4 |

## E    ADDITIONAL COMPARISONS

### E.1    COMPARISON WITH FOUNDHAND.

**Quantitative Comparison.** We conduct a quantitative comparison with FoundHand (Chen et al., 2025a) by finetuning HaMeR (Pavlakos et al., 2024) on 10k images generated by FoundHand and our method in Tab. 7. Since FoundHand does not provide detailed experiment setups about the domain transfer experiment from Re:InterHand (ReIH) to EpicKitchen in its paper, we follow the

Table 7: Quantitative comparison with FoundHand by finetuning HaMeR (Pavlakos et al., 2024) on 10k images generated by FoundHand and SesaHand (ours). The results of FoundHand are from its paper.

| FoundHand | PA-MPJPE ↓ | PA-MPVPE ↓ | SesaHand (ours) | PA-MPJPE ↓ | PA-MPVPE ↓ |
|---|---|---|---|---|---|
| Before finetune | 14.41 | 7.52 | Before finetune | 9.71 | 9.89 |
| After finetune | 13.68 $_{(-0.73)}$ | 6.29 $_{(-1.23)}$ | After finetune | 8.56 $_{(-1.15)}$ | 8.72 $_{(-1.17)}$ |

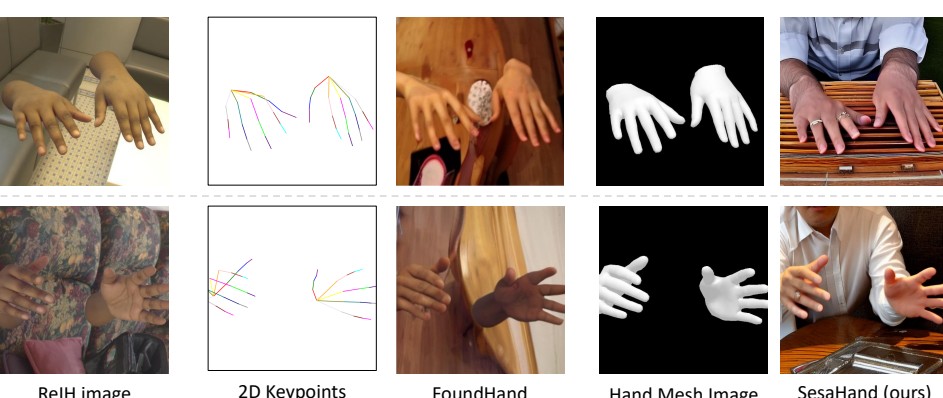

ReIH image        2D Keypoints        FoundHand        Hand Mesh Image        SesaHand (ours)

Figure 11: Comparison with FoundHand (Chen et al., 2025a) on hand image generation based on Re:InterHand (ReIH) (Moon et al., 2023) images. FoundHand shows floating hands in generated images whereas our method preserves proper hand-body alignment.

setting of HaMeR to train and report the evaluation results. We randomly sample 10k images from ReIH and render the corresponding hand mesh images. The text prompts are generated leveraging our CoT pipeline with semantic alignment. These hand mesh images and text prompts are used for hand image generation. We finetune HaMeR by these 10k images generated with our method. As shown in Tab. 7, our method can generate hand images to further improve HaMeR's 3D hand reconstruction performance, which both improve 3D joint and mesh prediction. On the contrary, FoundHand shows limited performance on 3D joint prediction, achieving 6.29 mm PA-MPVPE but only 13.68 mm PA-MPJPE after finetuning. Moreover, we show a comparison of the generated images of FoundHand and our method in Fig. 11. We use FoundHand to generate images from ReIH to EpicKitchen by using 2D keypoints from ReIH and target domain conditions from EpicKitchen. As observed, FoundHand shows floating hands in the generated hand images while our method produces hand images with hand-body alignment.

**Qualitative Comparison.** We compare our method with FoundHand, which supports gesture transfer, generating a hand image given a reference hand image and a target pose. We select a third-view image from MSCOCO and an ego-centric image from MOW (Cao et al., 2021). We obtain their corresponding 2D keypoints with Mediapipe, following the original implementation in FoundHand, and render the hand mesh image with the 3D ground-truth hand mesh provided by MOW. As shown in Fig. 12, the hand image generated by FoundHand has a hand-body misalignment issue, where the human body in the generated image is not coherent with the hand. Furthermore, the generation results are restricted by the reference images and cannot be easily controlled by text prompts, which limits its diversity. Moreover, FoundHand sticks to generating images in $256 \times 256$ resolution. Our method ensures the hand-body alignment in the generated images and can control the image setting with text prompts. We support a resolution of $512 \times 512$ with higher image quality. Notably, we can also generate a realistic hand image from an ego-centric view, while FoundHand fails to generate a plausible hand image.

## E.2 COMPARISON WITH HANDBOOSTER.

HandBooster (Xu et al., 2024) generates hand images conditioned on mesh images and additional normal images, which demonstrates its effectiveness for improving 3D hand reconstruction perfor-

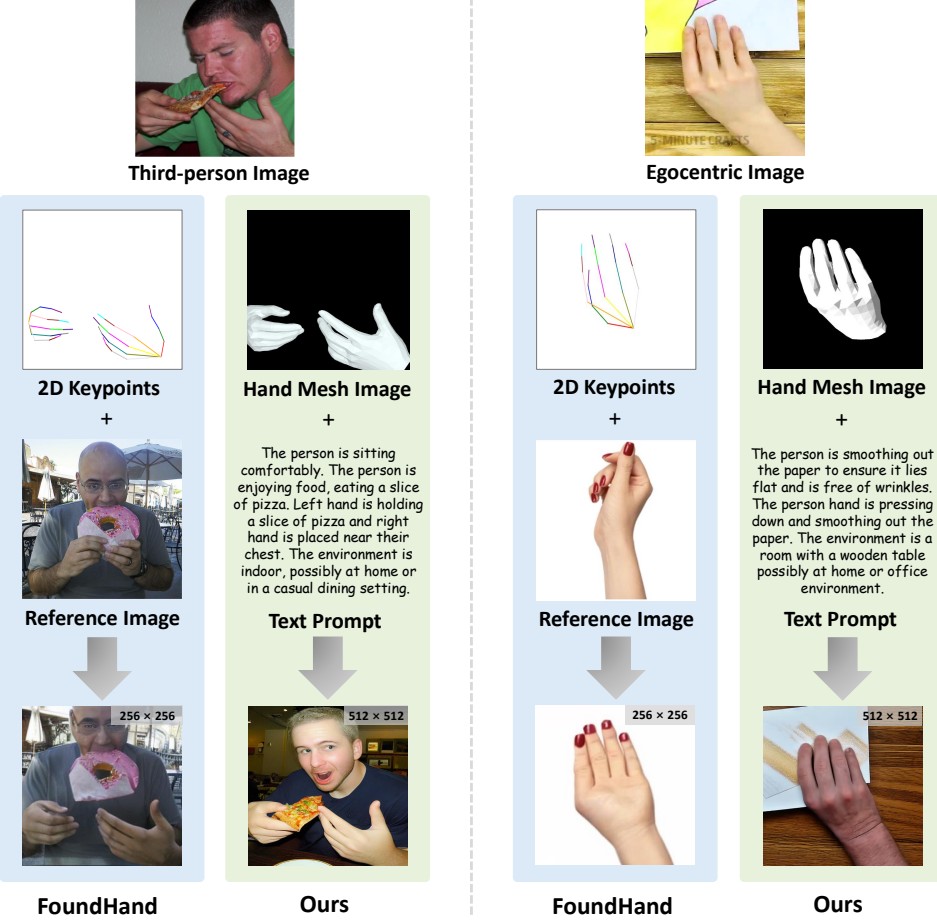

Figure 12: Comparison with FoundHand (Chen et al., 2025a). FoundHand generates a hand image in the target pose based on a reference image, which suffers from misalignment of the hand and body, as well as limited diversity. Our method achieves better hand-body alignment and can provide diverse settings through text prompts.

mance on lab environments (e.g., Dex-YCB (Chao et al., 2021) and HO3D (Hampali et al., 2020)). To evaluate its performance, we render the hand mesh and normal images from the Dex-YCB test set and conduct hand image generation with these conditions using HandBooster with its DexYCB s0 checkpoint. As shown in Fig. 13, the generated results show limited diversity in environment and texture, and the floating-hand issue remains noticeable. In contrast, our method generates well-aligned hand image with only a mesh image and a text prompt, even without training on the Dex-YCB dataset.

We further evaluate HandBooster by generating hand images with hand mesh and normal images from MSCOCO. As shown in Fig. 14 rows 1 and 2, HandBooster exhibits limited generalization ability for generating hand images conditioned on out-of-distribution mesh and normal conditions. Moreover, it also shows limited performance for generating small or interacting hands (see Fig. 14 rows 3 and 4), likely due to insufficient human priors and the restricted expressiveness of its training data. Our method can generate diverse in-the-wild hand images with plausible hand poses and hand-body alignment.

Moreover, we try to train HandBooster on the MSCOCO dataset to perform a comparison for improving 3D hand reconstruction as in Tab. 2. However, HandBooster is hard to converge when it is trained on images with complex and diverse environments, likely because it is trained from scratch without internet-scale human prior information. Additionally, HandBooster requires normal maps as an essential input condition, whereas our method does not require a normal map condition.

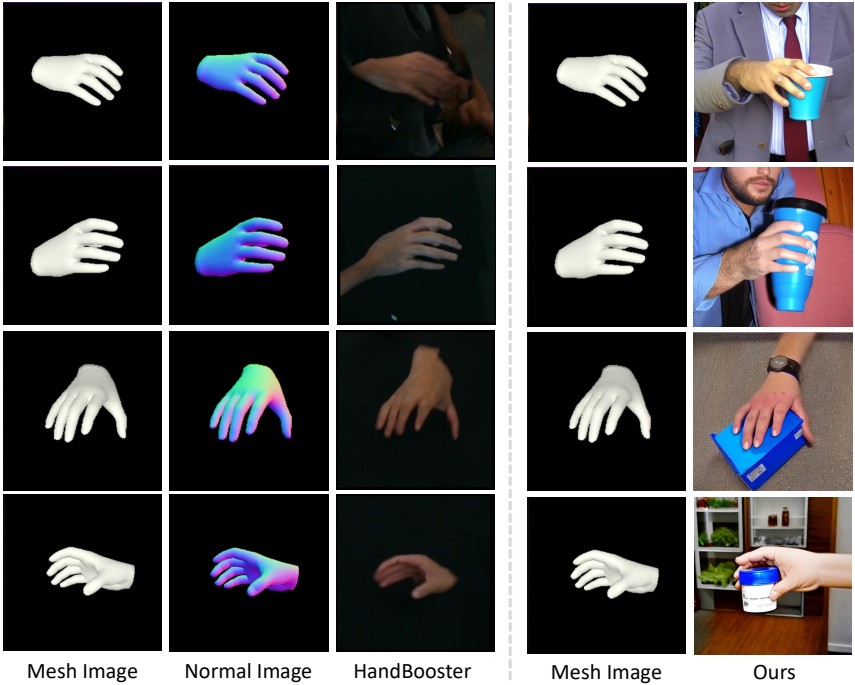

Figure 13: Qualitative comparison between HandBooster and our method with mesh and normal conditions from Dex-YCB. Normal image is required only by HandBooster.

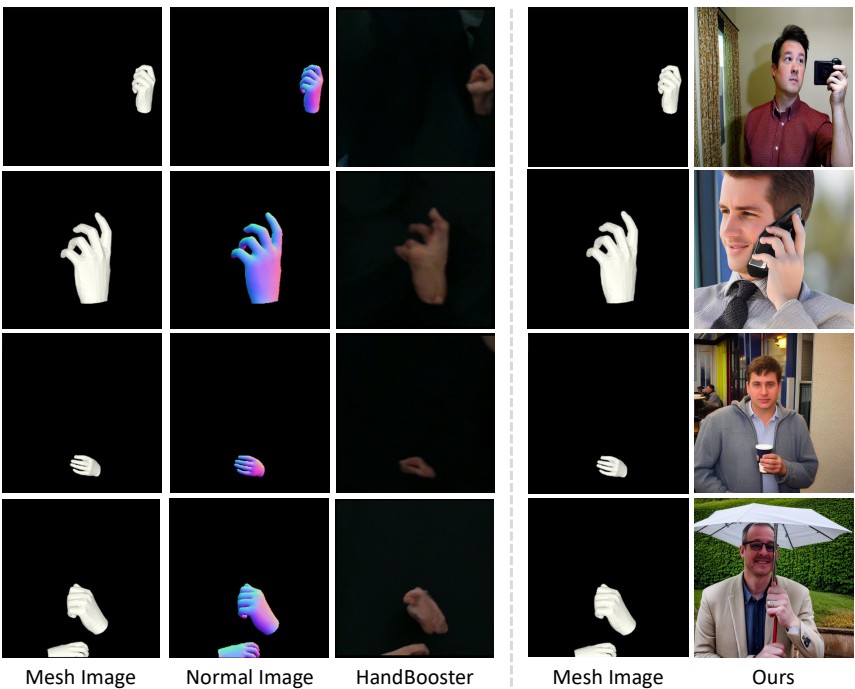

Figure 14: Qualitative comparison between HandBooster and our method with mesh and normal conditions from MSCOCO. Normal image is required only by HandBooster.

E.3    QUALITATIVE RESULTS OF HAND-OBJECT INTERACTION (HOI).

As shown in Fig. 15, our method can generate diverse hand images with hand-object interactions with text prompts and hand mesh renderings. Moreover, a challenging HOI of fingering a guitar is shown in Fig. 16. We randomly choose an image of fingering a guitar from Pexels. The text prompt is generated with our CoT pipeline and the hand mesh rendering is rendered by HaMeR for hand image generation.

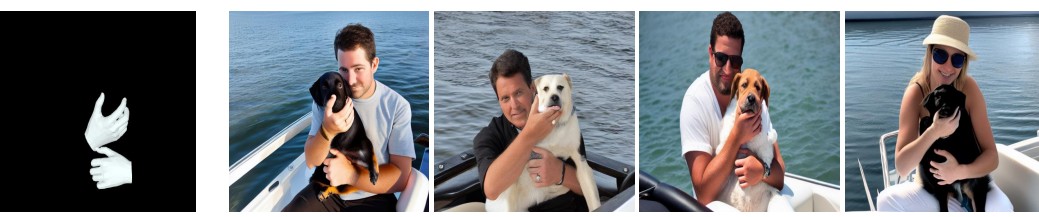

...The person is sitting with arms folded to hold the dog on a boat in the sea...

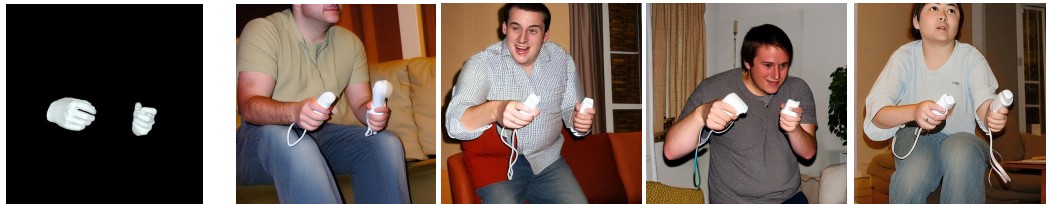

...The person is leaning forward slightly and playing a game on nintendo wii console...

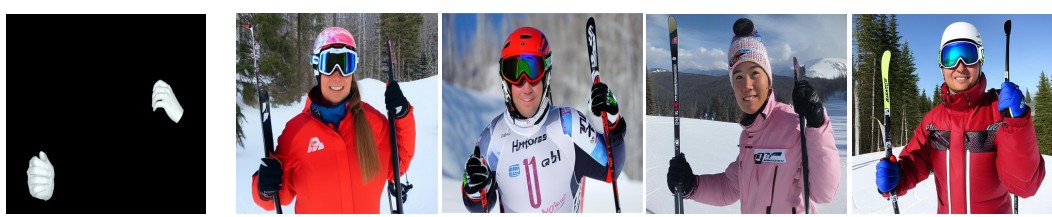

...The person is standing and preparing for skiing...

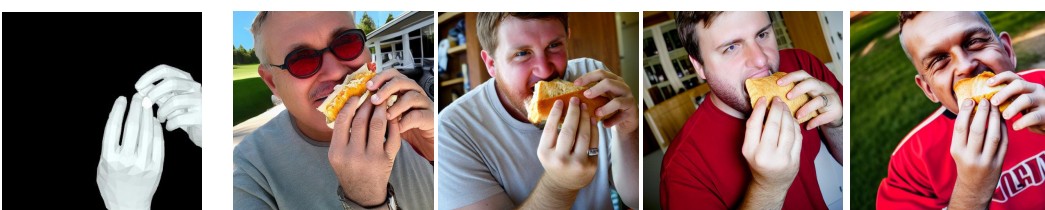

...The man is standing while both hands holding a sandwich with fingers wrapped around it...

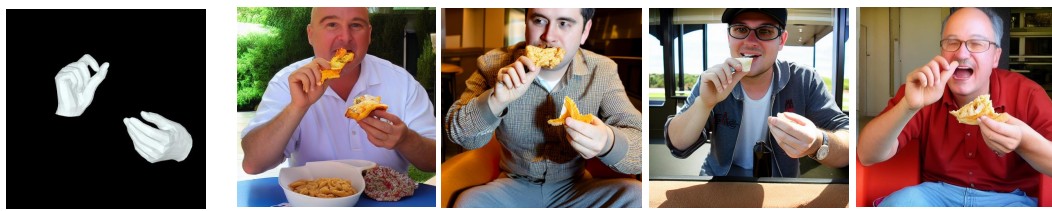

...The person is seated casually and holding a snack...

Figure 15: More qualitative results of the hand image generation with hand-object interactions.

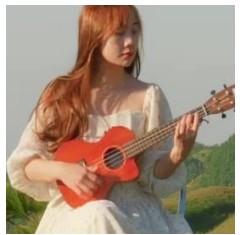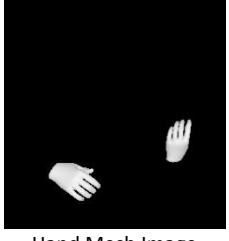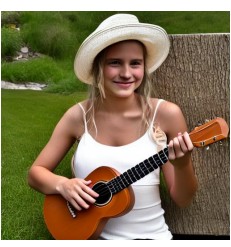

Real Image        Hand Mesh Image        Ours

…The person is seated with arms in playing position and playing on the ukulele…

Figure 16: Qualitative result on a challenging hand-object interaction (fingering a guitar). The real image is randomly chosen from Pexels. The text prompt is generated using our CoT pipeline and the hand mesh image is rendered by HaMeR.

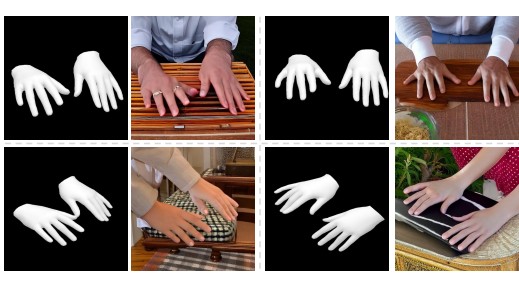 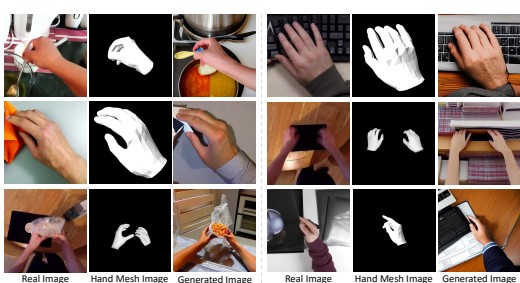

Figure 17: Qualitative results of hand image generation from different viewpoints.

Figure 18: Qualitative results of egocentric hand image generation using HOI4D (Liu et al., 2022) and MOW (Cao et al., 2021).

### E.4 QUALITATIVE RESULTS OF VIEWPOINT PERSPECTIVES AND EGOCENTRIC VIEWPOINTS.

We show more qualitative results of the hand image generation given hand mesh renderings from different viewpoint perspectives in Fig. 17 and egocentric viewpoints in Fig. 18.

### E.5 COMPARISON WITH CONTROLNET OPENPOSE FULL AND ATTENTIONHAND.

We provide more qualitative comparisons in Fig. 19, Fig. 20, and Fig. 21. Since AttentionHand has not released its checkpoint yet, we train its model using its official code for two months. However, due to the slow refinement process, the model still does not perform well. Note that the qualitative results presented in the main paper are taken from its paper. We further compare our method with ControlNet v1.1 Openpose Full, which enables image generation controlled by Openpose (Cao et al., 2019) body, hand, and face keypoints. As shown in Fig. 19, Openpose sometimes fails to detect the hand, leading to undesired generation results. Moreover, missing fingers occur in the generated images by ControlNet Openpose Full, even when hand keypoints are accurately detected. Additionally, the human body pose in the generated image is constrained by the detected pose from the input image, which limits the diversity of hand image generation. In contrast, our method can generate realistic and well-aligned hand images in various hand poses and shapes.

## F THE USE OF LARGE LANGUAGE MODELS

In this work, large language models were used only for language polishing and minor editing. All research ideas, methods, and experiments were carried out by human authors.

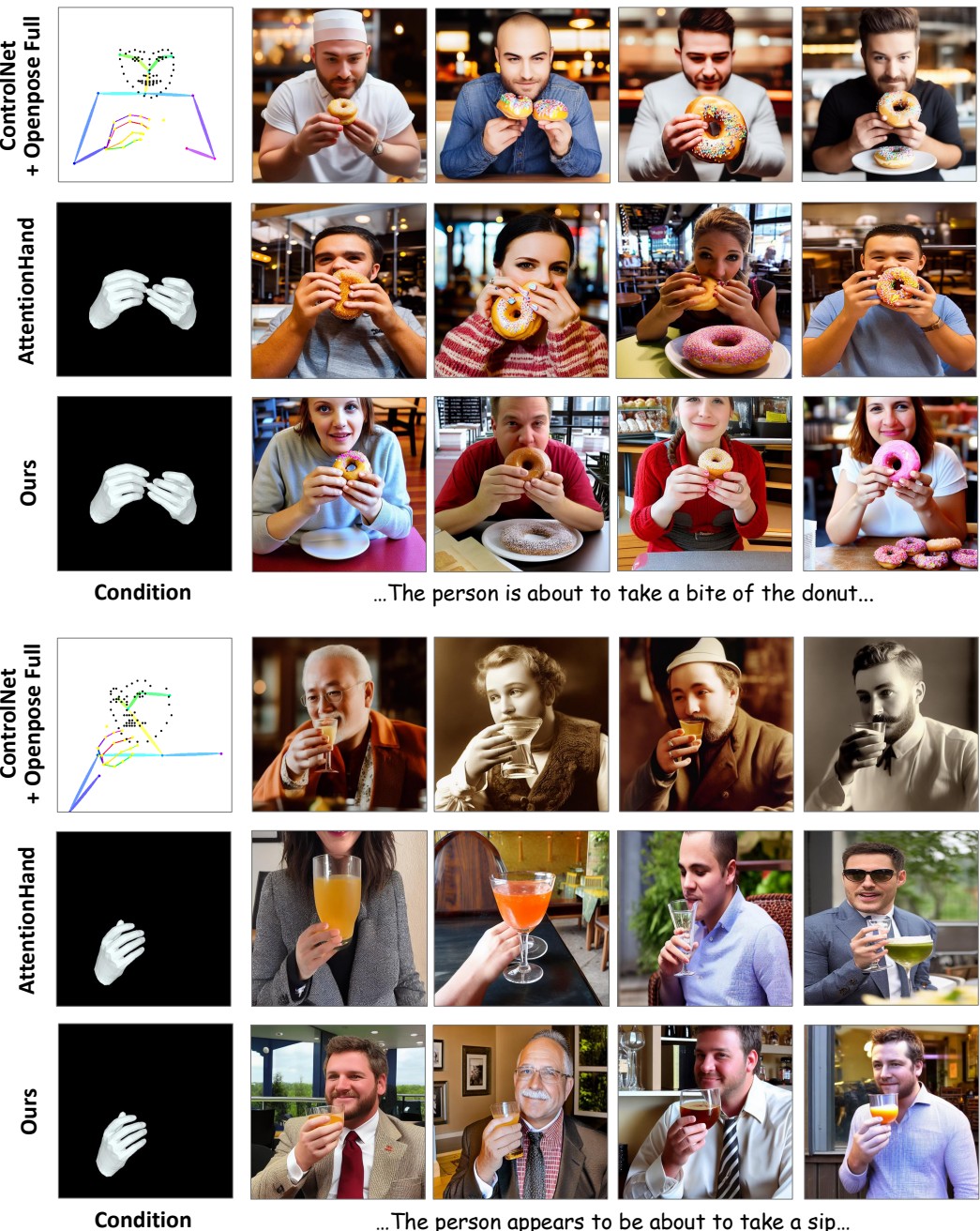

Figure 19: Qualitative comparison with ControlNet + Openpose Full and AttentionHand on conditional hand image generation. ControlNet + Openpose Full uses full-body keypoints detected by Openpose as input, while AttentionHand and our method take the hand mesh image as input. Our method demonstrates better generation performance with both semantic and structural alignment.

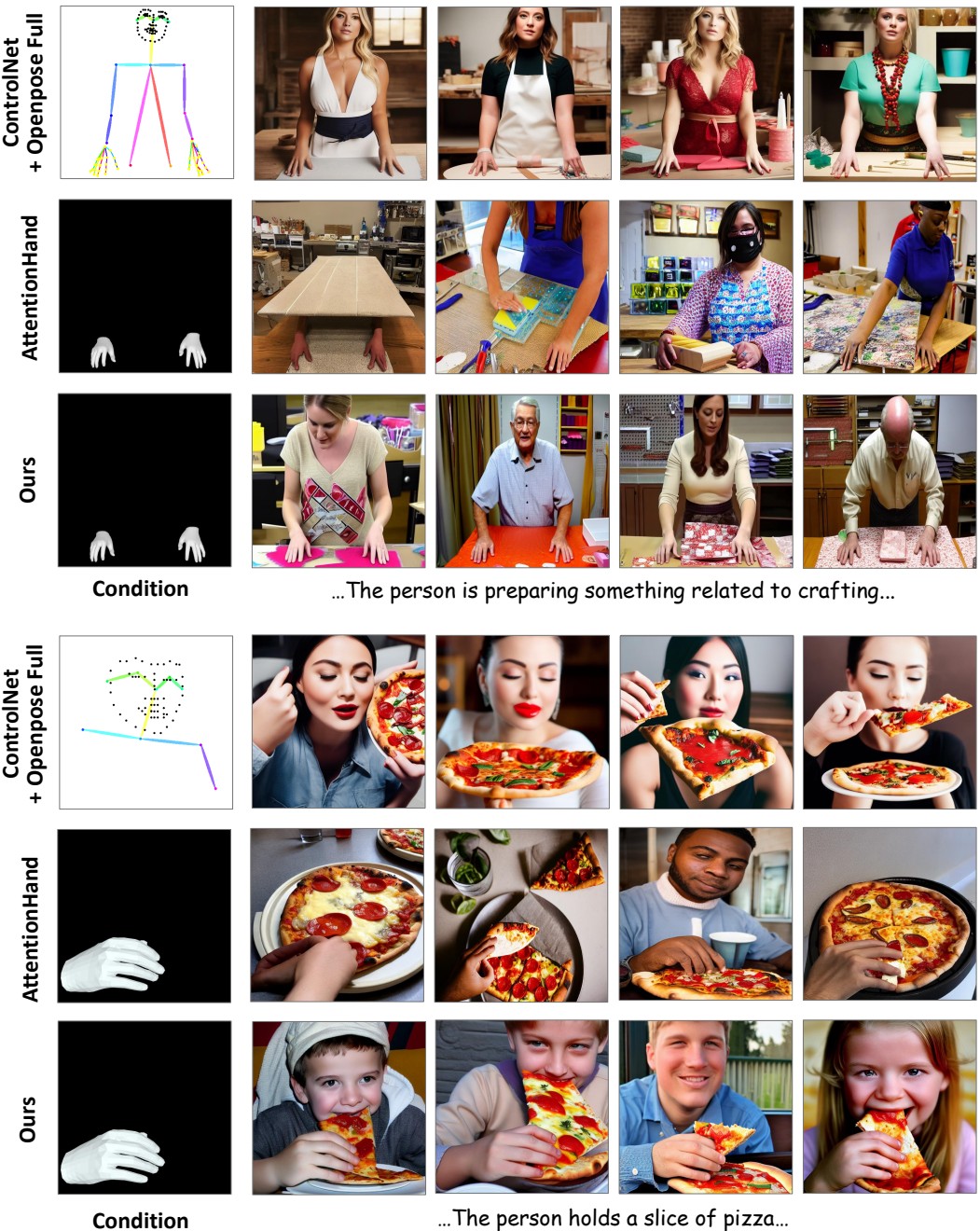

Figure 20: More qualitative comparison with ControlNet + Openpose Full and AttentionHand on conditional hand image generation.

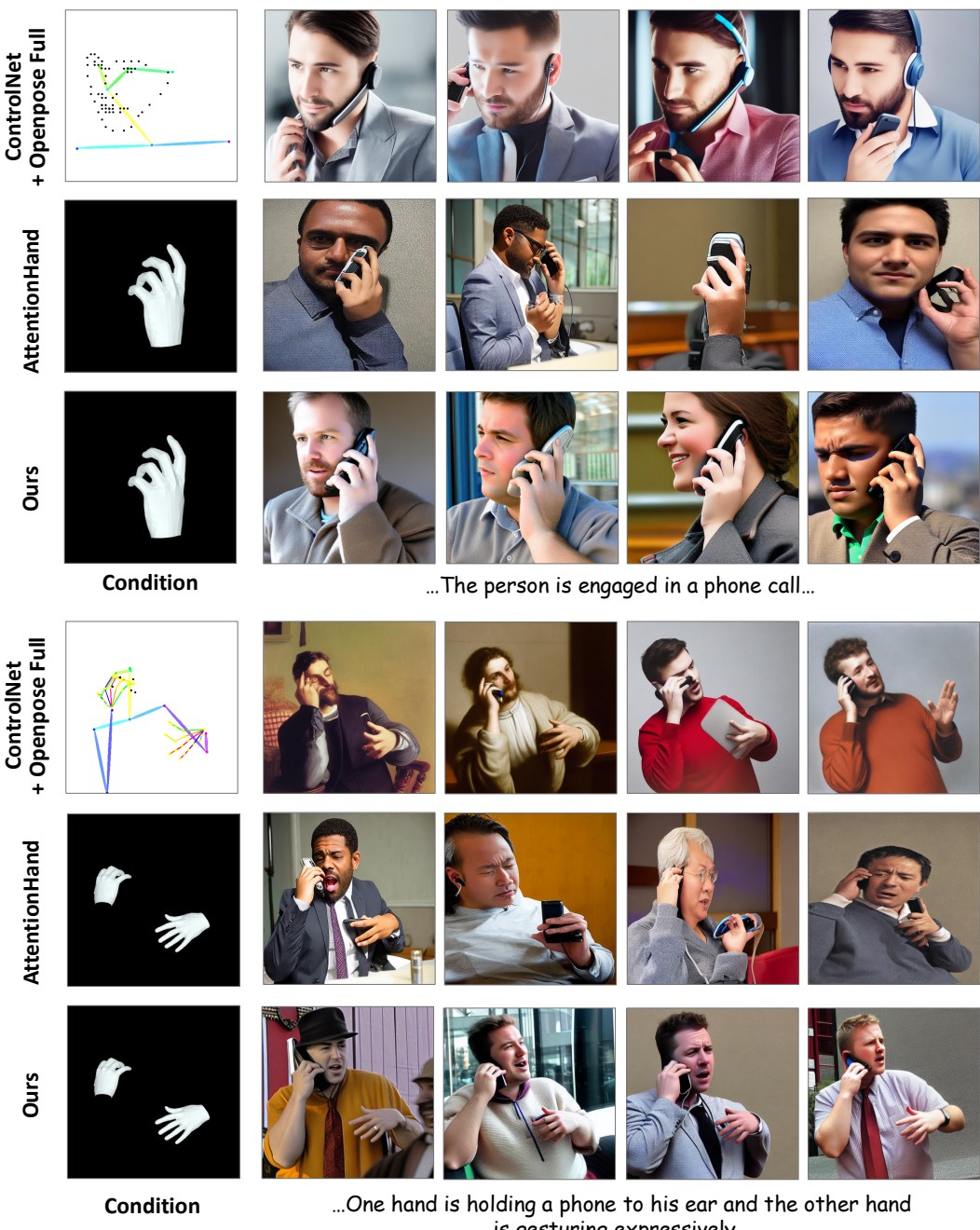

Figure 21: More qualitative comparison with ControlNet + Openpose Full and AttentionHand on conditional hand image generation.

