# OpenReview forum: "SesaHand: Enhancing 3D Hand Reconstruction via Controllable Generation with Semantic and Structural Alignment"
_ICLR.cc/2026/Conference — ICLR 2026 Poster_

### Official Review · Reviewer_CFW1 · 2025-10-31

**Soundness:** 3
**Presentation:** 4
**Contribution:** 3
**Rating:** 6
**Confidence:** 2

**Summary:**

This paper presents a mesh-controlled hand-image generation model, trained by a well-prepared dataset extracted through Chain-of-Thought (COT), and enhanced by multi-scale structural information and text-guided hand structure attention. Experiments demonstrate its superior performance in generation quality and improvement to the 3D hand reconstruction task, compared to SOTA methods.

**Strengths:**

1. This paper is in a good writing style.
2. The algorithm design, including image semantics extraction and two structural alignment modules, is reasonable. They are suited to the hand-image generation task.
3. The results show promising performance in both generation quality and benefiting hand reconstruction.

**Weaknesses:**

### Major Weakness
- The metrics reported in Table 1 are not complete compared to AttentionHand [1], including MSE-2D, MSE-3D, and user preference.
- In Table 2, HandBooster [2] can be compared, as it also uses mesh-controlled generated images to enhance 3D hand reconstruction.
- In evaluation, what's the text prompt input to the ControlNet/Stable Diffusion in SesaHand? Do they align with AttentionHand? This is important to ensure a fair comparison.
- In Table 2, for the re-implemented results of InterWild, why does only **RRVE on HIC** differ so much from the official one (24.20 vs. 21.35)? Yet the other metrics are all the same. Not sure if this is correct or not.

### Minor Weakness
- In the section of related work, ``Hand Image Synthesis`` and ``Diffusion-Based Hand Image Generation`` can be merged into one paragraph, as they are closely related.


[1] AttentionHand: Text-driven Controllable Hand Image Generation for 3D Hand Reconstruction in the Wild, ECCV 2024.
[2] HandBooster: Boosting 3d hand-mesh reconstruction by conditional synthesis and sampling of hand-object interactions, CVPR 2024.

**Questions:**

Although this task uses hand mesh to control image generation, I'm still wondering, can the model generate hand images without any hand mesh, just guided by the text? Will the SesaHand still perform better than previous methods in this case?

---

> ### Author Response · Authors · 2025-11-24
> **Response to Reviewer CFW1**
>
> Thank you for acknowledging SesaHand’s algorithm design, which is suited to the hand-image generation task, and giving positive feedback. We would like to provide further clarifications as follows:
>
> **W1: Missing metrics**
>
> We supplement our MSE-2D, MSE-3D and user preference results on Tab. 1 for your reference.
>
> **W2: Comparison with HandBooster**
>
> We are still running on it and will update the HandBooster results once it is complete.
>
> **W3: Evaluation**
>
> Our semantic alignment strategy is not applied during evaluation for a fair comparison and the text prompts in SesaHand are aligned with AttentionHand. As shown in Tab. 1 and Fig. 5, our quantitative and qualitative results demonstrate superior performance compared with previous methods under a fair comparison.
>
> **W4: Re-implementation of InterWild**
>
> During our re-implementation of InterWild on HIC evaluation, we found that AttentionHand reports its results using two different checkpoints: MPVPE and MRRPE are obtained from the InterWild checkpoint trained on IH26M (H+M) + MSCOCO + ReInterHand (Ego_cameras), while RRVE is reported using the checkpoint trained on IH26M (H+M) + MSCOCO + ReInterHand (Mugsy_cameras). In our paper, we adopt the InterWild checkpoint trained on IH26M (H+M) + MSCOCO + ReInterHand (Ego_cameras) for all reported results. We can achieve a better RRVE on the checkpoint (IH26M (H+M) + MSCOCO + ReInterHand (Mugsy_cameras)) of 20.40 compared with 21.10 of AttentionHand.
>
> **MW1: Related Works**
>
> Thank you for the suggestion! We have merged the two sections into a single paragraph to improve clarity and readability.
>
> **Q1: Generation without Hand Mesh Condition**
>
> Our model can generate hand images with only text prompts by retraining the model without the hand mesh images. However, we pursue a different goal from it. Our target is to generate hand images aligned with the labels so that these pair-wise data can be used for training 3D hand reconstruction models. Therefore, we mainly compared with controllable image generation methods. Although we mention some text-only hand-image generation methods in the related works section, their generated hand images can not be used for training due to the absence of corresponding labels.
>
> If you require more information, we are willing to supplement additional details during the discussion period. Thanks for your time and effort!

---

> ### Author Response · Authors · 2025-12-02
> **Update on Comparison with HandBooster**
>
> **W2: Comparison with HandBooster (Update)**
>
> We would like to further provide our experimental results on HandBooster. Firstly, we evaluate HandBooster using its DexYCB s0 checkpoint with mesh and normal conditions from Dex-YCB. HandBooster relies on normal maps as an essential input condition in addition to mesh images, while our method requires only mesh images and text prompts. We observe that the generated hand images from HandBooster show limited diversity in environments and textures, and the floating-hand issue remains noticeable (see Appendix E.6 Fig. 20), whereas our method generates diverse hand images with plausible hand-body alignment. Furthermore, its generalizability to out-of-distribution mesh conditions is limited, while our method can generate well-aligned hand images under such conditions (see Appendix E.6 Fig. 20 - 21).
>
> Secondly, we try to train HandBooster on the MSCOCO dataset to perform a comparison for improving 3D hand reconstruction as in Tab. 2. However, HandBooster is hard to converge when it is trained on images with complex and diverse environments, likely because it is trained from scratch without internet-scale human prior information. Additionally, since HandBooster requires normal maps while our method does not require a normal map condition, we consider that including HandBooster in Tab. 2 would not be a fair comparison.
>
> For more examples and detailed analysis, please refer to Appendix E.6. Thank you for your time and patience!

---

### Official Review · Reviewer_j2h5 · 2025-10-31

**Soundness:** 3
**Presentation:** 3
**Contribution:** 2
**Rating:** 6
**Confidence:** 2

**Summary:**

This paper proposes SesaHand, a framework for generating hand images conditioned on 3D hand parameters, achieving enhanced semantic and structural alignment in controllable hand image generation compared to existing methods. The synthesized images can be utilized for training 3D hand reconstruction models. The technical contributions are twofold. First, for semantic alignment, the framework employs chain-of-thought inference to generate effective human-centric image captions (used as inputs to the image diffusion model), thereby improving the semantic conditioning of the generated hand images. Second, for structural alignment, it introduces a hierarchical structural fusion mechanism to integrate hand structure information, along with a hand structure attention enhancement module. Experimental results demonstrate that the proposed method produces higher-quality hand images and yields better performance when used to train hand estimation models than the existing baselines.

**Strengths:**

**(1) Good presentation quality**

The paper is well written overall, and the text is easy to follow.

**(2) Good experimental results**

The proposed method achieves strong results in both (1) controlled hand image generation and (2) hand pose estimation (when training an estimation model on the generated hand images). However, I have a few questions regarding the comparison settings (see the weaknesses section below).

**(3) Good analysis to justify model design choices**

Although the proposed method is relatively simple and straightforward, it provides good analysis to justify the design choices. For example, I appreciate the prompt influence analysis used to validate the proposed text prompt extraction strategy.

**Weaknesses:**

**(1) Questionable omission of FoundHand in quantitative comparisons**

I believe FoundHand is one of the most relevant baseline works to this paper. Although it is discussed in the related work section and qualitative comparisons are provided in the supplementary material, I wonder why this baseline is omitted from the main quantitative experiments for both hand image generation and hand pose estimation. Since the code is publicly available, was there any specific reason why this comparison could not be performed?

**(2) Questionable experimental setting**

In the experiments, when explaining the suboptimal KID performance of this method compared to AttentionHand, it is mentioned that “Although our KID score slightly underperforms AttentionHand, this may be due to the random sampling strategy in the KID calculation.” However, this statement raises concerns about the whole experimental rigor. In related fields (e.g., diffusion-based motion generation [1]), it is standard practice to run N = 10 experiments and report the mean metric values to mitigate the influence of such randomness. Have you also considered this?

[1] Tevet et al., Human Motion Diffusion Model, ICLR 2023.

**(3) Novelty**

The proposed modules are not highly novel, as they largely combine existing techniques for enhancing the controllability of diffusion-based generative models. However, I consider this limitation relatively minor.

**Questions:**

Please refer to the weaknesses (1) and (2) above.

---

> ### Author Response · Authors · 2025-11-24
> **Response to Reviewer j2h5**
>
> Thank you for acknowledging SesaHand’s justification of our model design choices and good experimental results, and giving positive feedback. We address your concerns as follows:
>
> **W1: Comparison with FoundHand**
>
> FoundHand achieves controllable hand image generation through domain transfer, which requires 2D keypoints and reference images as conditioning inputs for structural and semantic controls. The reference image defines all elements in the generated image except the hand. It differs from our setting, where text prompts and hand mesh renderings are inputs. Therefore, for fairness, we compare with text-driven, hand mesh rendering-conditioned generation methods without reference image input in Tab. 1. To further address your concern, we provide additional quantitative comparisons with FoundHand in Appendix Sec. E1.1. Our method can achieve observable improvements on both 3D joint and mesh prediction performance compared with FoundHand (see Appendix Sec. E1.1 Tab. 7).
>
> **W2: Experimental Setting**
>
> We use clean-fid [i] to compute the KID score, which is a standard metric on image generation evaluation, and runs N=10 and max_subset_size = 100 to report the score. Since AttentionHand does not provide detailed information about its KID score calculation, we provide this explanation in the paper. To better address your concern, we further provide the standard deviations of our KID scores: KID (0.00553 ± 0.0015) and KID-H (0.00718 ± 0.0011) and the 95% confidence intervals are: KID (0.00553 ± 0.0029) and KID-H (0.00718 ± 0.0022).
>
> [i] On Aliased Resizing and Surprising Subtleties in GAN Evaluation. CVPR 2022.
>
> **W3: Novelty**
>
> Pre-trained Stable Diffusion (SD) models contain rich human prior information from large amounts of training data. Leveraging these priors to generate plausible hand images has great potential to enhance 3D hand reconstruction from a data augmentation perspective. However, existing methods that utilize the SD model for hand image generation still suffer from semantic and structural misalignments (e.g., unrelated environmental details that distract generation and hand-body misalignment). Our work is designed to conquer these challenges. For semantic alignment, we propose a CoT reasoning pipeline to generate human-centric contexts, which guides the SD model to focus on generating human-related regions and avoid unrelated artifacts. For structural alignment, we propose a hierarchical structural fusion to enhance global hand-body alignment and a hand structure enhancement method to highlight local hand regions. With our proposed strategies, our model generates semantic and structural-aligned hand images that improve 3D hand reconstruction.
>
> If you require further clarification, we are willing to supplement more details during the discussion period. Thanks for your time and effort!

---

### Official Review · Reviewer_46de · 2025-11-01

**Soundness:** 3
**Presentation:** 2
**Contribution:** 2
**Rating:** 4
**Confidence:** 4

**Summary:**

The paper proposes SesaHand, a controllable hand-image generation framework that aims to improve downstream 3D hand reconstruction by enforcing both semantic and structural alignment. On the semantic side, the method uses a CoT pipeline to distill human-behavior semantics from VLM captions. On the structural side, it aggregates multi-resolution self-attention maps via hierarchical fusion and adds a lightweight bias in cross-attention to emphasize hand regions. Built on ControlNet-conditioned diffusion with hand-mesh images, the model generates diverse, human-centric images and mitigates common synthetic-data issues such as “floating hands” and hand–scene misalignment.

**Strengths:**

1. The CoT-driven semantic extraction and the hierarchical structural fusion are well motivated and technically coherent; together they target complementary failure modes (semantic drift vs. structural misalignment).
2. The supplemental comparisons against commercial models indicate task-specific controllability and suggest that the proposed design adapts well to hand-centric scenarios where generic generators struggle.

**Weaknesses:**

1. The perceptual quality of generated images appears limited: several qualitative examples show unnatural finger articulation and color artifacts. Relative to SOTA hand rendering methods [i-iii] (acknowledging different goals and toolchains), the realism gap remains noticeable.
2. Section 4.2 does not provide sufficient detail for the reconstruction setup. The paper does not specify the number of synthetic images used, training schedule, the proportion of generated vs. original data, or whether models are trained from scratch or fine-tuned. These omissions hinder reproducibility and attribution of gains.
3. Efficiency is not quantified. The paper does not report module-wise runtime (CoT captioning, attention fusion, diffusion sampling), end-to-end throughput, training/inference cost, or memory usage. Given that efficiency is a practical criterion for synthetic data generation, the absence of such measurements weakens the empirical case.
4. The discussion of hand–object interaction synthesis is limited. The paper does not address how to generate more natural interaction images with plausible contact, occlusion handling, and object pose consistency—factors that matter for reconstruction in interactive scenes.

[i] Chen, Zhaoxi, et al. "URhand: Universal relightable hands." CVPR. 2024.

[ii] Potamias, Rolandos Alexandros, et al. "Handy: Towards a high fidelity 3d hand shape and appearance model." CVPR. 2023.

[iii] Iwase, Shun, et al. "Relightablehands: Efficient neural relighting of articulated hand models." CVPR. 2023.

**Questions:**

The following questions are based on the weaknesses discussed above; please refer to that section.

---

> ### Author Response · Authors · 2025-11-24
> **Response to Reviewer 46de**
>
> Thank you for recognizing SesaHand as a highly motivated and technically coherent hand-image generation framework for generating diverse human-centric images. We would like to address your concerns as follows:
>
> **W1: Perceptual Quality Relative to Hand Rendering Methods**
>
> The mentioned hand rendering methods aim to achieve 3D hand rendering with realistic hand textures. However, they require additional texture maps or environmental maps as inputs, or train an extra appearance parametric model with hand textures to achieve realistic rendering. They show floating hands and have semantic gaps between the hand and its surrounding environment. How to effectively leverage the renderings to enhance 3D hand reconstruction performance is under investigation. As the reviewer noted, our approach differs in both objectives and toolchains. Our goal is to facilitate the 3D hand reconstruction performance leveraging the generated hand images through our semantic and structural alignment strategy. We can achieve realistic and diverse hand image generation directly from text prompts and hand mesh renderings, which is more flexible and scalable.
>
> **W2: Reconstruction Setup**
>
> We have provided our implementation details in Appendix Sec. D, including our training schedule and details. 3D hand reconstruction models are fine-tuned, and the proportion of generated and original data is 1:5.
>
> **W3: Efficiency**
>
> Thanks for pointing this out! We provide additional module-wise runtime analysis and memory usage in Sec 4.3. Training details are provided in Appendix Sec. D.
>
> **W4: Hand-Object Interaction**
>
> The Stable Diffusion (SD) model has learned strong human and hand object interaction priors from large-scale training data, but it remains limited in controllability for reliable and high-quality hand image generation. Our method inherits the hand-object interaction priors from the SD model and enhances semantic and structural alignment during controllable generative model training, which implicitly improves the generation of hand images with hand-object interactions. Although we do not explicitly incorporate object information, we still demonstrate the ability to generate natural hand-object interaction images (see Appendix Sec. E.4) supported by the strong human priors learned within the pretrained generative model and our semantic and structural alignment strategies.
>
>
> Should you require more information, we are willing to provide further details during the discussion phase. Thanks for your time and effort!

---

### Official Review · Reviewer_bUzN · 2025-11-03

**Soundness:** 3
**Presentation:** 3
**Contribution:** 2
**Rating:** 4
**Confidence:** 5

**Summary:**

This paper presents a pipeline to generate synthetic hand images to improve 3D hand reconstruction. It addresses:
(1) Semantic alignment using COT to extract "human behavior semantics" for text prompts.
(2) Structural alignment using hierarchical fusion for hand-body coherence and an attention bias to focus on hand regions.
Experiments show that this generated data improves 3D hand reconstruction on in-the-wild datasets.

**Strengths:**

1. The problem statement of improving hand image generation quality to improve hand estimation task is sound. Data scarcity and fidelity is a key problems and current SOTA image generation models still fail to generate good hands occasionally.
2. This work shows using COT to improve the caption alignment of VLM and attention bias improves hand image generation compared with some prior works. Thereby, improving 3D hand reconstruction from synthetic images.
3. Writing and presentation clarity are good, and the paper is well-organized.

**Weaknesses:**

1. Though the paper focuses on improving semantic and hand structural alignment, objects in HOI alignment seem to be neglected; most examples only show simple cases of hands holding stuff. This work would be stronger if a more challenging and comprehensive study of HOI generation were demonstrated.
2. The examples lack diversity in style and viewpoint perspectives. It's difficult to judge how robust the proposed method is.
3. Comparison with the latest works, like FoundHand, is under investigation. See questions below.

**Questions:**

1. In Fig. 7 w/ SE+SF+AE, the coffee-holding hand in the first example has a "broken" finger, and the small left hand in the second example has only 4 fingers. Despite all the improvement modules, the model still seems to struggle with generating plausible hand structures. What is the cause of this, and how do authors plan to address this next?
2. Why not consider an explicit object condition, like object mask, for HOI alignment? How does the proposed method perform for more challenging HOIs, like fingering a guitar?
3. FoundHand shows in their Domain Transfer experiment that finetuning HaMeR with 10k generated images from ReInterHand to EpicKitchen improves 3D reconstruction accuracy. How does SesaHand compare with this baseline?
4. Most examples are shown in 3rd third-person view. The authors claim that their method can also generate an egocentric view, but Fig.10 only shows one sample of an egocentric view. Why not include a comprehensive experimental comparison on some egocentric hand data, e.g. HOI4D, ARCTIC, EpicKitchen, etc? Also, FoundHand shows many examples of egocentric generation in their results.

---

> ### Author Response · Authors · 2025-11-24
> **Response to Reviewer bUzN (1/2)**
>
> Thank you for acknowledging SesaHand’s semantic and structure alignment strategies to generate hand images for improving 3D hand reconstruction performance. We would like to address your concerns as follows:
>
> **W1: HOI Alignment**
>
> The pretrained Stable Diffusion (SD) model has learned strong priors about hand-object interactions from large-scale training data. Our method enhances the learning of the hand region and leverages semantic alignment to guide the model's attention to generate the hand region more effectively while inheriting the HOI priors from the pretrained SD model, which implicitly improves HOI learning. As shown in the paper (Fig. 5, Fig. 17-19), our method can generate high-quality hand images with hand-object interactions. To further address your concern, we provide additional challenging HOI examples in Appendix Sec. E.4 Fig. 13. We can observe that the generated hand images can naturally depict two hands interacting with various objects.
>
> **W2: Limited Diversity in Style and Viewpoint**
>
> Our method can customize the style of hand image generation given text prompts (e.g., background, texture, and gender configurations), offering greater flexibility and diversity compared with methods that only rely on 2D image-based conditions (e.g., skeleton, texture maps). As shown in Appendix Fig. 13, Figs. 17-19, the generated hand images show diversity in styles, including variety in background, texture, and gender. Moreover, our method can generate hand images given different viewpoint hand mesh renderings. To better address your concern, we provide additional examples in Appendix Sec. E.5 Fig. 15, where we collect hand mesh renderings from different viewpoints from the ReIH dataset and generate the corresponding hand images.
>
> **Q1: Plausible Hand Structure Generation**
>
> In Fig. 7, we present two challenging cases: the upper one has occlusions, and the lower one has varying hand shape scales. It is a common issue that generative models still underperform in generating occluded and small hands, which may result in defects such as missing fingers. To address your concern, we further conduct a user study and report the user preference score with a 5-point Likert scale compared with previous methods in Tab. 1. The user preference results show that our method generates more plausible and realistic hand images compared with previous methods from human perceptual perspectives. To further address this issue, we plan to adopt a prioritized training strategy that emphasizes higher noise levels training by applying a higher sampling probability. It has been discovered that prioritizing the training on higher noise levels yields better performance [i], since higher noise levels focus on generating images’ structural information while lower noise levels focus on refining details [ii]. We believe applying this training strategy together with our structural alignment modules can enhance the hand structure learning to mitigate the missing fingers issue.
>
> [i] Perception prioritized training of diffusion models. CVPR 2022.
>
> [ii] FADE: Frequency-Aware Diffusion Model Factorization for Video Editing. CVPR 2025.
>
> **Q2: Explicit Object Condition**
>
> Text-driven HOI image generation can more efficiently and effectively generate large amounts of data for training. Although an explicit object condition can specify object information, it hinders efficient hand image generation since more efforts are needed to align the hand mesh and object for plausible and compatible results. Moreover, the explicit object condition like object masks could have artifacts (e.g., imperfect segmentation of occluded objects), which may result in undesired performance degradation. To better address your concern, we supplement a challenging HOI of fingering a guitar in Appendix Fig. 14. As observed, our method can generate a plausible hand-object interaction image without an explicit object condition.
>
> **Q3: Comparison with FoundHand**
>
> Since FoundHand does not provide detailed experiment setups about their domain transfer experiment in its paper, we follow the setting of HaMeR to train and report the evaluation results. We generate the text prompts with our CoT pipeline and render the hand mesh images from Re:InterHand to generate 10k images. We finetune HaMeR using our generated images and report our results compared with FoundHand in Appendix Sec. E.1.1 Tab. 7. Our method can generate hand images that further improve HaMeR's 3D hand reconstruction performance, where both 3D joint and mesh predictions show improvements. On the contrary, FoundHand shows limited performance on 3D joint prediction, achieving 6.29 mm PA-MPVPE but only 13.68 mm PA-MPJPE after finetuning.

---

> ### Author Response · Authors · 2025-11-24
> **Response to Reviewer bUzN (2/2)**
>
> **Q4: Egocentric Generation**
>
> Our method is designed to generate semantic and structural-aligned hand images that are useful for improving 3D hand reconstruction performance. To ensure a fair comparison with previous methods, we use the same training set that does not include egocentric hand datasets. Although our method is not trained on egocentric datasets, it can generalize to certain egocentric scenarios. To better address your concern, we supplement additional examples on egocentric hand generation in Appendix Sec. E.4, Fig. 16 with HOI4D and MOW. While FoundHand achieves controllable egocentric hand image generation through domain transfer, it requires a reference image as a target-domain condition for semantic control (see Appendix Fig. 11), which specifies all elements of the generated image except the hand or hand pose. In contrast, our method does not rely on reference images and uses text prompts to provide a more flexible form of semantic control for generation.
>
> If you have further questions, we are happy to provide further clarifications during the discussion phase. Thanks for your time and effort!

---

> ### Comment · Reviewer_bUzN · 2025-11-24
>
> Thanks for the comprehensive response and additional insights the authors provided. I believe the fundamental limitation of this work is the dependence on the pretrained SD, which is probably the main culprit causing the occasional low fidelity of the generated hand. Nevertheless, the authors have addressed most of my concerns well. I have decided to increase my rating of this paper.

---

### Author Response · Authors · 2025-12-02
**Summary of the Reviews and Responses**

Dear Area Chairs,

We express our sincere gratitude for your devotion to the ICLR 2026 conference. To facilitate your evaluation, we would like to provide a brief summary of each reviewer’s comments and our responses as follows.

**Strengths**

Reviewers recognize that our proposed **CoT-driven semantic extraction** and the **hierarchical structural fusion** for hand image generation to improve the hand estimation task are **sound, well motivated and technically coherent**. They also appreciate our **good analysis to justify the design choices** (prompt influence analysis), and the **algorithm design** (image semantics extraction and two structural alignment modules) **is suited to the hand-image generation task**. They acknowledge that our proposed method **achieves strong results in both controlled hand image generation and hand pose estimation**.

Reviewer bUzN has **raised the rating of our paper to 6 on 25 Nov 2025** for our comprehensive response and additional insights we provided during the discussion period.

**Concerns and Responses**

- Reviewer bUzN concerns HOI generation, diversity in style and viewpoint perspectives, comparison with FoundHand, and egocentric view generation. We provided detailed clarification of our implicit HOI learning inheriting human priors from the Stable Diffusion model, and our semantic and structural alignment strategies. We also provided more samples with style and viewpoint diversity, quantitative and qualitative comparisons with FoundHand, and the hand image generation ability from egocentric views.

- Reviewer 46de concerns the perceptual quality relative to hand rendering methods, experimental setup, image generation efficiency, and HOI generation discussion. We provided our clarification on the differences in objectives and toolchains compared with hand rendering methods. We have added additional reconstruction setups and efficiency analysis. We also provided a thorough discussion on natural HOI generation with hand-object interaction priors and enhanced semantic and structural information.

- Reviewer j2h5 had questions regarding the comparison with FoundHand and the KID metric calculation. We provided our clarification of the difference between FoundHand and our method, and more experimental results compared with FoundHand. We also provided explanations of our KID calculation and included the 95% confidence intervals to confirm statistical meaningfulness.

- Reviewer CFW1 concerns the additional metrics, comparison with HandBooster, text prompt consistency in evaluation, and the RRVE score on HIC. To address his concern, we added additional metric scores and provided a detailed comparison and analysis with HandBooster. We also provided clarification regarding the text prompt consistency and the RRVE score on HIC.

The detailed response can be found in the official comment to reviewers. We have also revised the manuscript with the modified parts highlighted in blue. We hope that the above summary can facilitate your decision-making process and make ICLR 2026 another success. Thanks again for your time and effort!

Best,

ICLR Submission 6601 Authors

---

### Meta-Review · Area_Chair_ZeHc · 2026-01-07

**Summary:**

This work received 4, 4, 6, 6 scores, i.e. two borderline accept and two borderline reject recommendations from reviewers. One of the main concerns was to compare FoundHand, which the authors additionally compared in the rebuttals. Low perceptual qualities and missing results in egocentric views were the other concerns. The authors provided more clarities and results in the rebuttals. Overall, the rebuttals have addressed the reviewer concerns and ACs recommend acceptance.

**Reviewer Concerns:**

The same as above.

**Reviewer Scores:**

This work received 4, 4, 6, 6 scores, and the author rebuttals have addressed the issues.

---

### Decision · Program_Chairs · 2026-01-26

Accept (Poster)